# *Vibrio parahaemolyticus* becomes lethal to post-larvae shrimp via acquiring novel virulence factors

Shuang Liu,[1,2,3] Wei Wang,[1,2] Tianchang Jia,[1,2] Lusheng Xin,[1,2,3] Ting-ting Xu,[1,2,3] Chong Wang,[1,2] Guosi Xie,[1,2,3] Kun Luo,[1,2] Jun Li,[4] Jie Kong,[1,2,3] Qingli Zhang[1,2,3]

**ABSTRACT** Translucent post-larvae disease (TPD), caused by *Vibrio parahaemolyticus* ($Vp_{TPD}$), has become an emerging shrimp disease, affecting more than 70%–80% of coastal shrimp nurseries in China in spring 2020. Here, we investigated the key virulence factors of $Vp_{TPD}$ by analyzing protein fragments, related genomic information, as well as experimental challenge tests. After investigating the toxic effects of purified protein fragments with different molecular weights (MWs) from $Vp_{TPD}$, we found that only the protein fragments with MW >100 kDa showed similar lethality to live $Vp_{TPD}$ in the experimental challenge test using post-larvae shrimp. Meanwhile, similar histopathological changes exhibiting in the hepatopancreas and midgut of the diseased individuals were observed in the post-larvae shrimp challenged with either bacterial protein fragments (MW >100 kDa) or live $Vp_{TPD}$. Based on sodium dodecyl sulfate-polyacrylamide gel electrophoresis (SDS-PAGE) and mass spectrometry analyses, two novel proteins, *Vibrio* high virulent protein (VHVP)-1 and VHVP-2, were identified as the candidates of key virulence factors to cause TPD. Moreover, VHVP-1 and VHVP-2 were found to be encoded by two genes (*vhvp-1* and *vhvp-2*) tandemly located on a 187,791-bp plasmid and were predicted to depend on the same promoter following a comparative genomic analysis. Further epidemiological investigation and challenge test indicated that the *V. parahaemolyticus* isolate carrying only the *vhvp-1* gene and lacking *vhvp-2* gene could not cause mortality of experimental *Penaeus vannamei* post-larvae. The mutant (Δvhvp-2) by deleting *vhvp-2* gene could only cause 4.92% of accumulative mortality of post-larvae that is similar to the non-$Vp_{TPD}$ *Vibrio* strain. Additionally, the complemented strains, Δvhvp-2/pBT3-vhvp-2 and Δvhvp-2/pwtCas9-vhvp-2, showed similar virulence to the wild-type $Vp_{TPD}$. The results demonstrated that *V. parahaemolyticus* becomes lethal to post-larval shrimp by acquiring the VHVP-2 virulence factor. This study sheds light on further investigations of the pathogenic mechanism of $Vp_{TPD}$ and the development of strategies for early diagnosis of TPD in shrimp hatcheries.

**IMPORTANCE** As a severe emerging shrimp disease, TPD has heavily impacted the shrimp aquaculture industry and resulted in serious economic losses in China since spring 2020. This study aimed to identify the key virulent factors and related genes of the $Vp_{TPD}$, for a better understanding of its pathogenicity of the novel highly lethal infectious pathogen, as well as its molecular epidemiological characteristics in China. The present study revealed that a novel protein, *Vibrio* high virulent protein-2 (MW >100 kDa), is responsible to the lethal virulence of *V. parahaemolyticus* to shrimp post-larvae. The results are essential for effectively diagnosing and monitoring novel pathogenic bacteria, like $Vp_{TPD}$, in aquaculture shrimps and would be beneficial to the fisheries department in early warning of $Vp_{TPD}$ emergence and developing prevention strategies to reduce economic losses due to severe outbreaks of TPD. Elucidation of the key virulence genes and genomics of $Vp_{TPD}$ could also provide valuable information on the evolution and ecology of this emerging pathogen in aquaculture environments.

Address correspondence to Qingli Zhang, zhangql@ysfri.ac.cn.

The authors declare no conflict of interest.

See the funding table on p. 17.

**KEYWORDS** translucent post-larvae disease (TPD), *Vibrio parahaemolyticus*-causing TPD (*Vp*TPD), shrimp, novel virulence factor, *Vibrio* high virulent protein (VHVP)

From late 2019 to early 2020, a new shrimp disease called translucent post-larvae disease (TPD) or glass post-larvae disease appeared in the southern coastal provinces of China. TPD became more and more prevalent in shrimp post-larvae, causing collapse of 70%–80% coastal shrimp nurseries in China in the spring of 2020 (1–3). A highly virulent *Vibrio parahaemolyticus* strain (*Vp*-JS20200428004-2) was identified as the responsible pathogen for the infectious TPD and was provisionally named as *V. parahaemolyticus* causing TPD, or *Vp*TPD (4). *Vp*TPD was highly lethal in particular to post-larvae at 4–7 days old (PL4–PL7). The cumulative mortality of the infected post-larvae could reach up to 100% in 3 days in a typical disease case. The infected shrimp post-larvae exhibited typical clinical syndromes, such as pale or colorless hepatopancreas and empty digestive tract, which made the diseased individuals to become transparent and translucent; therefore, these diseased individuals were named "translucent post-larvae" or "glass post-larvae" by local farmers (4).

*Vp*TPD infection in the *Penaeus vannamei* post-larvae could cause obvious histopathological changes that are similar to some degree to those of acute hepatopancreatic necrosis disease (AHPND). The epithelial cells of hepatopancreatic tubules and midgut were necrotic and sloughed off. A large number of colonized bacteria could be observed in hepatopancreas and midgut under microscope (4). Whereas, the toxicity of *Vp*TPD (vp-HL-202005) to the post-larvae of *P. vannamei* was about 1,000 times higher than that of the *V. parahaemolyticus* strain causing AHPND (3).

Until 2023, the prevalence of TPD was still common in *P. vannamei* nurseries and farms in the coastal provinces of China. Even though some antibiotics were reported to be able to kill or inhibit *Vp*TPD, the demand of antibiotic-free shrimp production prompted the high preference of biosecurity measures, including early detection and disinfection treatment to prevent the occurrence and prevalence of the TPD. Therefore, there is an urgent need to investigate the key virulence factor of *Vp*TPD for developing effective diagnostic techniques and further prevention strategies of TPD.

In the present study, we first carried out investigations for the key virulent proteins with different molecular weights that contribute to the pathogenicity of *Vp*TPD to *P. vannamei* post-larvae via experimental challenge tests. Then, the virulent protein fragments as potential virulence factors of *Vp*TPD were characterized by mass spectrometry and genome sequencing. Meanwhile, we also investigated the presence of *Vibrio* high virulent protein (VHVP) virulence factor in different *Vibrio* isolates as well as its occurrence in the TPD cases in different shrimp farms from different geographical areas of China. The results of our current study should shed insights into the molecular pathogenic mechanisms of *Vp*TPD in *P. vannamei* post-larvae.

## RESULTS

### Inactivation of *Vp*TPD

We first tested the effect of thermal inactivation and ultrasonic disruption to inactivate *Vp*TPD. The culture of *Vp*TPD ($7.1 \times 10E8$ CFU/mL) was treated for inactivation by different combinations of two methods, ultrasonic disruption (U) and heating (H) at 65°C for 45 min. The lysate protein extract obtained by ultrasonic disruption of *Vp*TPD (>100 kDa) was inactivated too. The viability of inactivated *Vp*TPD was tested by inoculating different treatments onto agar plates, and no bacteria grew on the plates from the treatment groups of *Vp*TPD + U and H, *Vp*TPD + U and H (>100 kDa), and *Vp*TPD + U (>100 kDa), which indicated that *Vp*TPD could be effectively inactivated via various combination methods of sonication and pasteurization in the present study.

## Pathogenicity of the candidate virulence factors of $Vp_{TPD}$ determined by the challenge test

The cumulative mortality rates of challenged *P. vannamei* post-larvae with both live $Vp_{TPD}$ and its protein fractions with different molecular weights are shown in Fig. 1b. During a 40-h experimental period, no death occurred in the negative control (NC) group; however, dead post-larval shrimps in the group challenged with $7.1 \times 10^5$ CFU/mL of $Vp_{TPD}$ were observed at 8 h, and the mortality reached 100% after 24 h of challenge. The mortality of shrimps in the group of $Vp_{TPD}$ + U (>100 kDa) began at 16 h post of challenge and then reached 90% after 32 h. In contrast, the cumulative mortality of post-larvae in all the groups of $Vp_{TPD}$ + U (50–100 kDa), $Vp_{TPD}$ + U (30–50 kDa), $Vp_{TPD}$ + U (10–30 kDa), and $Vp_{TPD}$ + U (<10 kDa) did not exceed 10% even after 32 h of challenge (Fig. 1b). The results indicated that only $Vp_{TPD}$ + U (>100 kDa) proteins showed a similar virulent effects to *P. vannamei* post-larvae as live $Vp_{TPD}$, which means the efficient virulence factors of $Vp_{TPD}$ should be in the fraction (MW >100 kDa) of the lysate protein extract by ultrasonic disruption of $Vp_{TPD}$ + U.

## Histopathological analysis of samples from different challenged groups

Histopathological examination revealed severe necrosis and sloughing of epithelial cells in both hepatopancreatic tubules and midgut of the infected post-larvae with live $Vp_{TPD}$ at 24 h post challenge (Fig. 1c). In the group challenged with $Vp_{TPD}$ + U (>100 kDa), mild necrosis and sloughing of epithelial cells were observed in both hepatopancreatic tubules and midgut at 16 and 24 h post challenge, and severe necrosis and sloughing of epithelial cells occurred at 32 and 40 h post challenge (Fig. 1d); the most severe histopathological changes were seen in the midgut at 32 h post challenge, and severe necrosis of epithelial cells causes epithelial cells to fall off the basement membrane of the midgut and scatter into the cavity of the midgut (Fig. 1d). In contrast, there were no obvious histopathological changes in the hepatopancreatic tubules and midgut of the post-larval individuals from the control group (Fig. 1c).

## Identification of $Vp_{TPD}$ virulence factors by SDS-PAGE and mass spectrometry analysis

To screen the candidate virulence factors from this ultrasonic disruption lysate protein extract with molecular weight >100 kDa in $Vp_{TPD}$, the SDS-PAGE analysis showed three bands representing three major proteins in the $Vp_{TPD}$ + U portion (MW >100 kDa) (Fig. 2a). The three bands, designated as $Vp_{TPD}$_4-2-1, $Vp_{TPD}$_4-2-2, $Vp_{TPD}$_4-2-3, were then excised from the gel and projected for further analysis by using a mass spectrometer and were identified as insecticidal toxin complex protein (GenBank: WP_269169668.1), virulence protein (GenBank: APX09935.1) (Fig. 2b and c), and aconitate hydratase B (GenBank: KIT24301.1), respectively. Finally, both $Vp_{TPD}$_4-2-1 and $Vp_{TPD}$_4-2-2 were selected as the candidate virulence factors I and II of $Vp_{TPD}$ for further analysis, and $Vp_{TPD}$_4-2-3 was excluded from subsequent analyses, as aconitate hydratase B is not a virulence protein according to previous reports.

## Genome sequencing and comparative genome analysis of $Vp_{TPD}$ and non-$Vp_{TPD}$ strains

To better understand the genetic information of the virulence factors of $Vp_{TPD}$, a comparative genome analysis of $Vp_{TPD}$ and non-$Vp_{TPD}$ strains was carried out, and sequencing results showed that the complete genome of $Vp_{TPD}$ consisted of two circular chromosomes (Fig. 3a) and three plasmids (Fig. 3c). The two circular chromosomes are 3,527,627 bp (chromosome 1) and 1,887,516 bp (chromosome 2) in length, respectively. The three plasmids of $Vp_{TPD}$ are 212,543 bp, 187,791 bp, and 60,506 bp, respectively. Whereas, the complete genome of the non-$Vp_{TPD}$ strain ($Vp_{1616}$) consisted of two circular chromosomes (Fig. 3b) and without plasmids. The two circular chromosomes

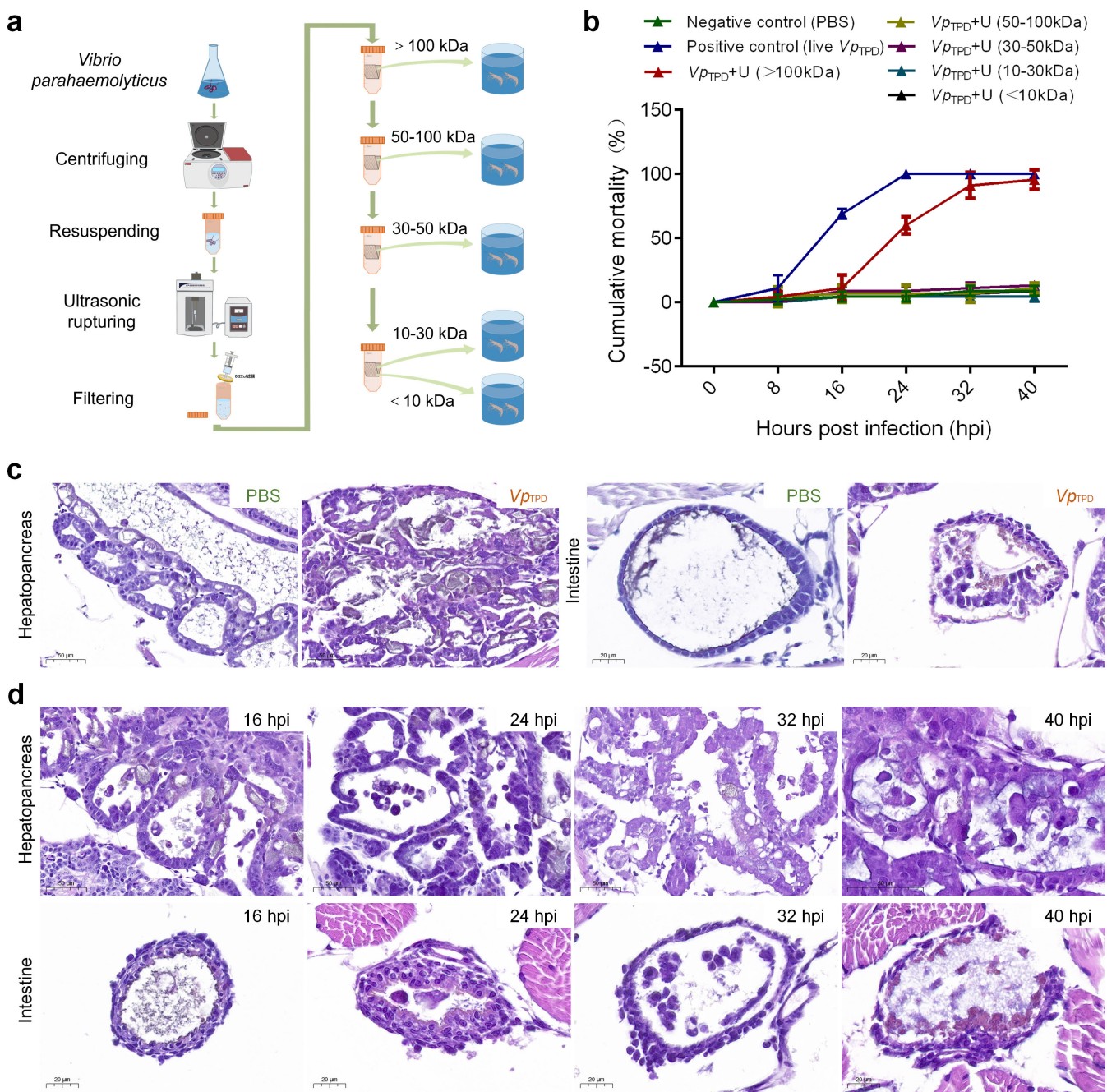

**FIG 1** Pathogenicity analysis of $Vp_{TPD}$ proteins of different molecular weights to *Penaeus vannamei* post-larvae. (a) Schematic of the protocols used to obtain $Vp_{TPD}$ proteins of different molecular weights. (b) Cumulative mortality of *P. vannamei* post-larvae induced by different molecular weights of $Vp_{TPD}$ proteins in the immersion challenge test. Each group contained three experimental tanks as three replicates. For each replicate, 15 shrimps were challenged by immersion with 1× PBS buffer (negative control), live $Vp_{TPD}$ (positive control), and the proteins of $Vp_{TPD}$ with different molecular weights ($Vp_{TPD}$ + U [>100 kDa], $Vp_{TPD}$ + U [50–100 kDa], $Vp_{TPD}$ + U [30–50 kDa], $Vp_{TPD}$ + U [10–30 kDa], $Vp_{TPD}$ + U[(<10 kDa]), respectively. Cumulative mortality of shrimp was shown as the mean and SD of three replicate data for each experimental group. For each replicate, healthy shrimps were immersed in a concentration of $7.1 \times 10^5$ CFU/mL live $Vp_{TPD}$ (infected group) or in a concentration of protein fractions with different molecular weights extracted from $7.1 \times 10^5$ CFU/mL $Vp_{TPD}$. (c) Histopathological photographs of hepatopancreas and intestine of *P. vannamei* post-larvae from the live $Vp_{TPD}$-challenged group (positive control) and 1× PBS-challenged group (negative control). (d) Histopathological photographs of hepatopancreas and intestine of *P. vannamei* post-larvae from $Vp_{TPD}$ + U (>100 kDa) challenged group at different time points post infection (including 8 hpi, 16 hpi, 24 hpi, 32 hpi, and 40 hpi).

are 3,288,162 bp (chromosome 1) and 1,923,178 bp (chromosome 2), respectively. The comparative analysis of genomic information between $Vp_{TPD}$ and $Vp_{1616}$ demonstrated

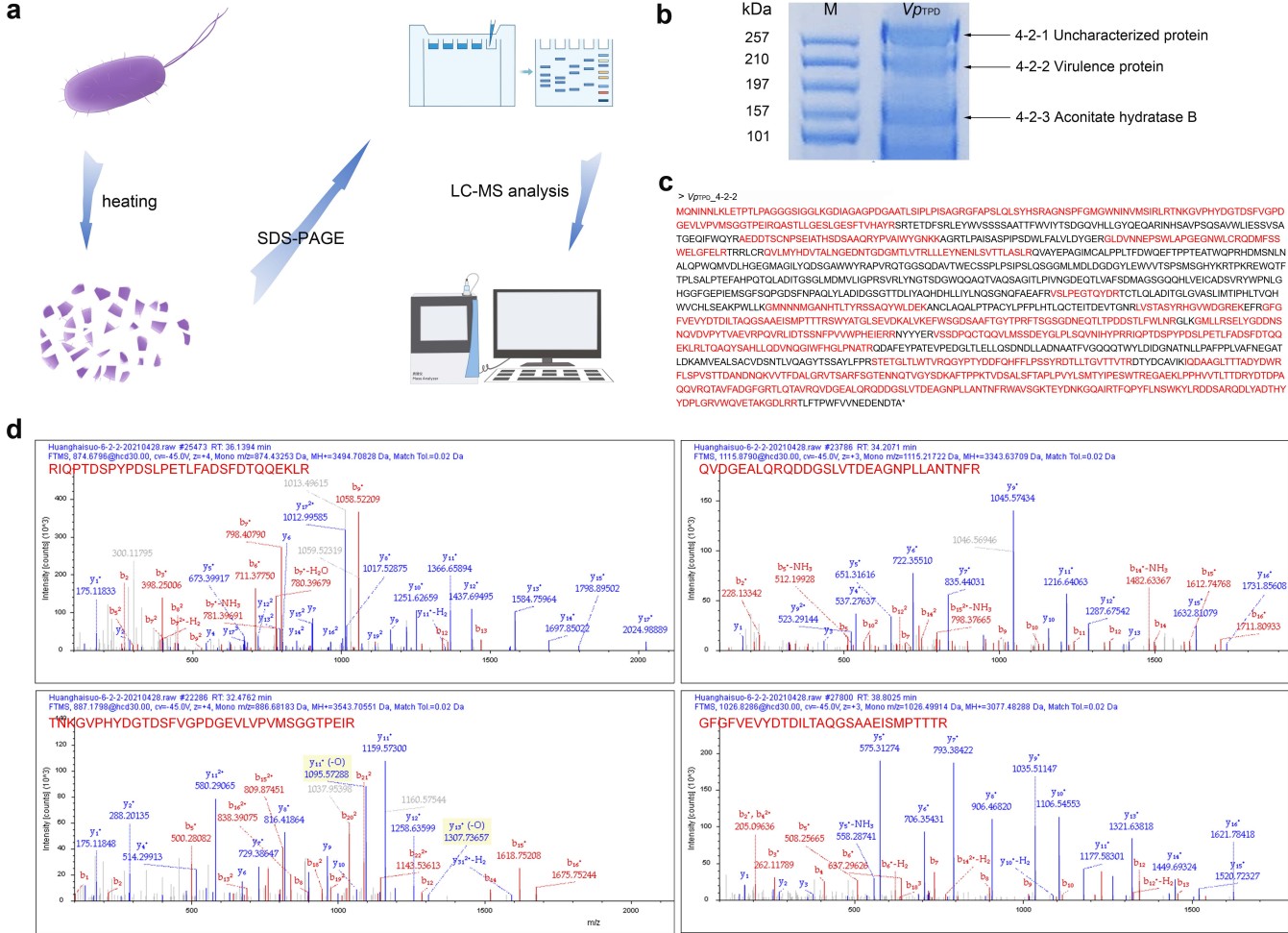

**FIG 2** SDS-PAGE and mass spectrometry analysis of $Vp_{TPD}$. (a) Schematic of protein sample analysis of $Vp_{TPD}$. (b) $Vp_{TPD}$ proteins revealed by SDS-PAGE electrophoresis. Lane 1, $Vp_{TPD}$; lane M, protein molecular weight marker (kDa). The major proteins in $Vp_{TPD}$ with molecular weights >100 kDa, $Vp_{TPD}$_4-2-1, $Vp_{TPD}$_4-2-2, and $Vp_{TPD}$_4-2-3, were identified as insecticidal toxin complex protein (GenBank: WP_269169668.1), virulence protein (GenBank: APX09935.1), and aconitate hydratase B (GenBank: KIT24301.1), respectively. (c) Identification of $Vp_{TPD}$ virulence proteins by matrix-assisted laser desorption ionization-time-of-flight mass spectrometry analysis. Sequences in red font are the peptide sequences identified by mass spectrometry. (d) Identification of $Vp_{TPD}$ proteins by matrix-assisted laser desorption ionization-time-of-flight mass spectrometry analysis. Part of the secondary mass spectrum of the sequence in red font.

that two putative virulent factor genes (GE005140 and GE005139) only presented in the $Vp_{TPD}$ but not in $Vp_{1616}$. According to the results of multiple sequence alignment using the online Blastx program on the NCBI web, the two putative virulent factor genes (GE005140 and GE005139) were found to encode the deduced candidate virulence factors I and II, which shared 100% and 99.49% amino acid sequences identity with the insecticidal toxin protein (GenBank: WP_269169668.1) and the virulence protein (GenBank: APX09935.1), respectively.

## Sequence characterization of the unique virulence factors of $Vp_{TPD}$

Based on the comparative genomic and mass spectrometry analysis, two putative virulent proteins of $Vp_{TPD}$_4-2-1 and $Vp_{TPD}$_4-2-2, which were encoded by GE005140 and GE005139 in $Vp_{TPD}$, respectively, were named as putative VHVP-1 and VHVP-2. The genes of GE005140 (*vhvp-1*) and GE005139 (*vhvp-2*) were found to be tandemly located on a 187,791-bp plasmid of the $Vp_{TPD}$ genome and are predicted to depend on the same promoter in the plasmid by using the classic bacterial sigma70 promoter recognition program. According to the open reading frame (ORF) finder analysis (https://

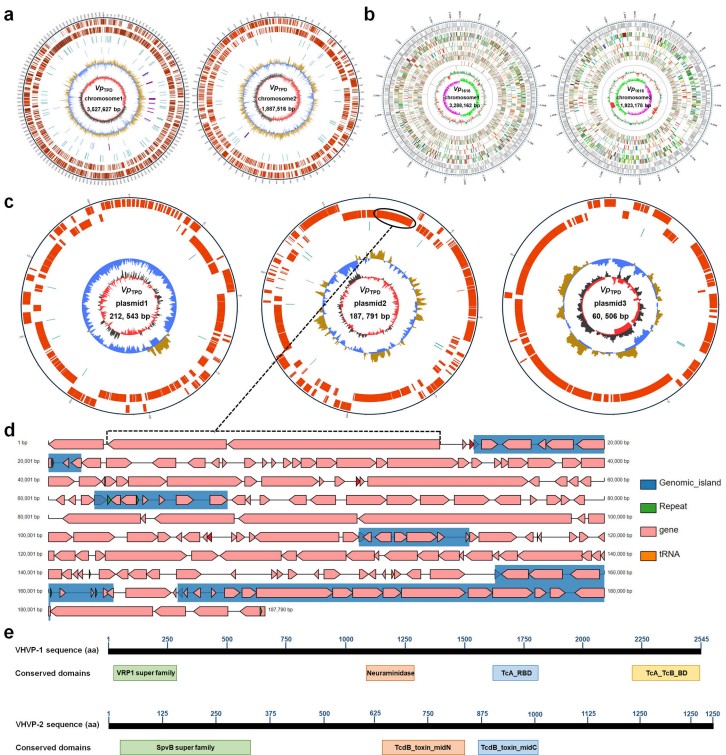

**FIG 3** Circular genome and plamid maps of chromosomes of $Vp_{TPD}$ and $Vp_{1616}$, and domain structure of gene *vhvp-1* and *vhvp-2* of $Vp_{TPD}$. (a) Circular genome maps of chromosome 1 and chromosome 2 of $Vp_{TPD}$. From inner to outer, the first circle represents the genomic length in 5 kb; the second and third circles represent the COG function category of the protein-coding sequence on the forward and reverse strands, respectively; the fourth circle represents the repetitive sequence; the fifth circle represents tRNA and rRNA, blue is tRNA, and purple is rRNA; the sixth circle represents the GC content; the innermost circle is the GC skew, dark gray represents the area where the G content is greater than C, and the red represents the area where the C content is greater than G. (b) Circular genome maps of chromosome 1 and chromosome 2 of $Vp_{1616}$. The outermost circle represents the positional coordinates of the genome sequence. From outside to inside, they are coding genes, gene function annotation results (including COG, KOG, eggNOG, KEGG, and GO database annotation results information), ncRNA, genome GC content, and genome GC skew value distribution. Circular genome maps of chromosome 1 and chromosome 2 of $Vp_{1616}$. The outermost circle represents the positional coordinates of the genome sequence. From outside to inside, they are coding genes, gene function annotation results (including COG, KOG, eggNOG, KEGG, and GO database annotation result information), ncRNA, genome GC content, and genome GC skew value distribution. (c) Circular maps of plasmid 1, plasmid 2, and plasmid 3 of $Vp_{TPD}$. From inner to outer, the first circle represents the genomic length in 5 kb; the second and third circles represent the COG function category of the protein-coding sequence on the forward and reverse strand, respectively; the fourth circle represents the repetitive sequence; the fifth circle represents tRNA and rRNA, where blue is tRNA, and purple is rRNA; the sixth circle represents the GC content; the innermost circle is the GC skew, dark gray represents the area where the G content is greater than C, and the red represents the area where the C content is greater than G. (d) Genome map of plasmid 2 (187,791 bp) of $Vp_{TPD}$. The black circle represents the position of the putative virulence genes of *vhvp-1* (GE005140) and *vhvp-2* (GE005139) in the genome. (e) The deduced conserved domain structure of the proteins encoded by *vhvp-1* and *vhvp-2* in plasmid 2. VRP1 super family, *Salmonella* virulence plasmid 28.1 kDa A protein; *Neuramin*, neuraminidase-like domain; *TcA*, TcA receptorbinding domain; *TcA_TcB_BD*, Tc toxin complex TcA C-terminal TcB-binding domain. *SpvB*, *Salmonella* virulence plasmid 65 kDa B protein; *TcdB*, bacterial insecticide toxin *TcdB*.

www.ncbi.nlm.nih.gov/orffinder/) of the *vhvp* genes, VHVP-1 was composed of 2,544 amino acid residues, with a predicted molecular mass of 283.37 kDa and a predicted pI of 4.69, and VHVP-2 was composed of 1,421 amino acid residues, with a predicted molecular mass of 161.34 kDa and a predicted pI of 4.63. Prediction by the online Conserved Domain Search Service (CD Search) in NCBI revealed that VHVP-1 possessed the conserved domains of Tc toxin complex TcA C-terminal TcB-binding domain (Pfam ID: CL39627), TcA receptor-binding domain (Pfam ID: CL139842.1), neuraminidase-like domain (Pfam ID: pfam18413), and *Salmonella* virulence plasmid 28.1 kDa A protein (Pfam ID: CL21676) (Fig. 3e). Meanwhile, VHVP-2 was found to contain the conserved domains of *Salmonella* virulence plasmid 65 kDa B protein (*SpvB*, Pfam ID: pfam03534), insecticide toxin *TcdB* middle/C-terminal region (Pfam ID: pfam12255), and insecticide toxin *TcdB* middle/N-terminal region domain (Pfam ID: cl13663) (Fig. 3e). VHVP-1 and VHVP-2 shared 45.16%–99.49% and 71.85%–100% overall sequence identity with other bacterial virulence factors, respectively.

## Detection of $Vp_{TPD}$ by PCR

In order to develop a PCR detection method for $Vp_{TPD}$, PCR primers were designed for targeting *vhvp-1* and *vhvp-2* genes (Fig. 4a; Table 3). Both DNA samples from the $Vp_{TPD}$ isolate and shrimp tissues suffered with TPD could be amplified and produced 362, 351, and 303 bp amplicons using the $Vp_{TPD}$-*vhvp-1*-F1/R1, $Vp_{TPD}$-*vhvp-2*-F1/R1, and $Vp_{TPD}$-*vhvp-2*-F2/*R2* primer sets, respectively (Fig. 4a). Specificity analysis of the primers was performed by using DNA samples from the non-$Vp_{TPD}$ strains, including *V. parahaemolyticus*-0421B, *Pseudoalteromonas flavipulchra* (CDM8), *V. parahaemolyticus* causing AHPND ($Vp_{AHPND}$, 20200610006-16), *V. alginolyticus* (20150606001-2), *V. harveyi* (20170902102-3), *V. owensii* (20150709001-2), and *V. campbellii* (20150606027-2). The results showed that no expected PCR products were amplified when using DNA from non-$Vp_{TPD}$ strains as templates, which indicated that the PCR primer sets are only specific for $Vp_{TPD}$ (Fig. 4b).

## Epidemiological analysis of $Vp_{TPD}$

A total number of 179 shrimp samples were collected from different shrimp farms in China. Field epidemiological investigations and laboratory histopathological analyses revealed that the TPD occurred in shrimp farms in Hebei, Shandong, Jiangsu, Hainan, and Xinjiang provinces (Fig. 4c). All DNA samples extracted from 179 shrimp samples were subjected to molecular epidemiological investigations using the specific $Vp_{TPD}$ PCR assay. The PCR results showed that the targeted *vhvp-1* (containing *VRP1*, neuraminidase, and *TcA* domains) and *vhvp-2* gene (containing *SpvB* and *TcdB* domains) could only be amplified in the shrimp samples with typical TPD cases but not from the healthy or non-TPD shrimp samples (Fig. 4c). In addition, the *V. parahaemolyticus* isolate 20211213002-3 that was isolated from shrimp farm in Hunan Province and carry only the *vhvp-1* gene but no *vhvp-2* gene could not cause mortality of experimental *P. vannamei* post-larvae in the challenge test (Fig. 5a). The results showed that the *vhvp-2* gene, rather than the *vhvp-1* gene, is the actual key virulence gene in the $Vp_{TPD}$.

## Confirmation of the key virulence factor of $Vp_{TPD}$

The nucleotide sequence of the coding sequences (CDS) of the *vhvp-2* gene shared 100% to 71.85% sequence identity with its homologs in *Vibrio campbellii, Vibrio parahaemolyticus, Photobacterium damselae, Vibrio owensii, Photobacterium iliopiscarium, Aliivibrio fischeri, Yersinia ruckeri,* and *Enterobacter asburiae*. To confirm the key virulence factor of $Vp_{TPD}$, an isogenic mutant of $Vp_{TPD}$ (Δvhvp-2) was constructed. And in the Δvhvp-2, including the entire insecticide toxin TcdB middle/N-terminal region domain, from the 106th amino acid residue of conserved domain of the *Salmonella* virulence plasmid 65 kDa B protein to the 79th residue of the conserved domain of the insecticide toxin TcdB middle/C-terminal region was successfully deleted. The lethal effects of Δvhvp-2

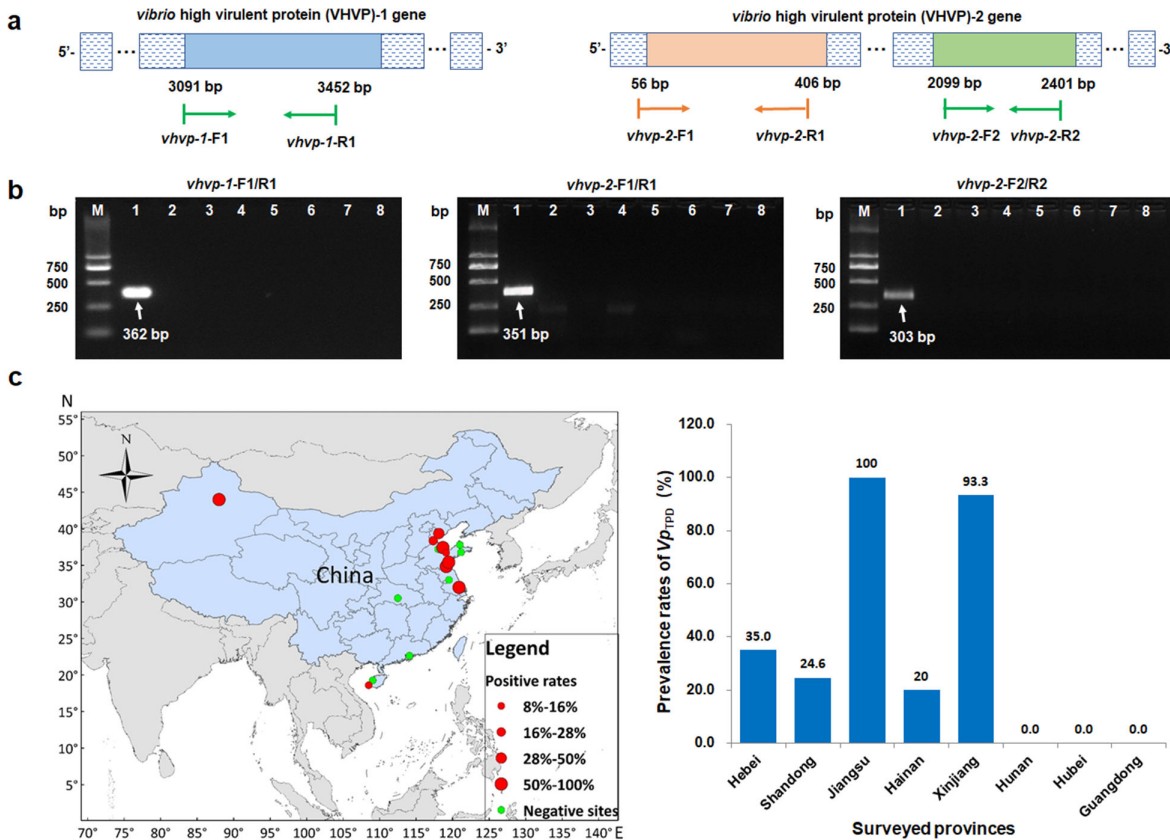

**FIG 4** Sites schematic of PCR primers of $Vp_{TPD}$ virulence factor genes and the molecular epidemiological analysis based on *vhvp* gene of $Vp_{TPD}$. (a) Schematic of the $Vp_{TPD}$ virulence factor (*vhvp*) gene *vhvp-1* and *vhvp-2* and the detection primers targeting the *vhvp* genes. (b) Electrophoretogram of molecular detection of the *vhvp* genes encoding the conserved domain of *TcdA* in *vhvp-1* gene, *SpvB* and *TcdB* in *vhvp-2* gene in different marine pathogens. Lane 1, $Vp_{TPD}$; lane 2, *Vibrio parahaemolyticus*-0421B; lane 3, *Pseudoalteromonas* (CDM8); lane 4, *Vibrio parahaemolyticus* causing AHPND (20200610006-16); lane 5, *Vibrio alginolyticus* (20150606001-2); lane 6, *Vibrio harveyi* (20170902102-3); lane 7, *Vibrio owensii* (20150709001-2); lane 8, *Vibrio campbellii* (20150606027-2); lane M, molecular weight marker (bp). (c) $Vp_{TPD}$ prevalence in different shrimp aquaculture regions with different prevalence rates (left). $Vp_{TPD}$ prevalence rates in different sampling province in the TPD epidemiological survey (right). Shrimp samples were collected from the shrimp farms in Hebei, Shandong, Jiangsu, Hainan, Xinjiang, Hunan, Hubei, and Guangdong provinces of China from April, 2020 to 2021. $Vp_{TPD}$ prevalence rates in the histogram only represent the positive detection rate of the *vhvp-2* gene in the collected samples and not the actual prevalence condition of the TPD in the local areas. The map in panel C was created using ArcGIS 10.4.

and $Vp_{TPD}$ to post-larval shrimp were compared by experimental challenge, and the results showed that at the same dose of pathogen, $Vp_{TPD}$ caused 81.89% mortality at 32 h post challenge, Δvhvp-2 caused 4.92% mortality, while the negative control caused no death (Fig. 5b). The cumulative mortality induced by $Vp_{TPD}$ was significantly different from that of Δvhvp-2 and NC. Furthermore, the mortality induced by NC was significantly lower than the two complement strains Δvhvp-2/pBT3-vhvp-2 and Δvhvp-2/pwtCas9-vhvp-2, and the wild-type $Vp_{TPD}$ (Fig. 5b). The results indicate that the protein of VHVP-2 is key to the pathogenic effect of $Vp_{TPD}$, and therefore it was considered as the key virulence factor of $Vp_{TPD}$.

## DISCUSSIONS

TPD, a new emerging disease mainly affecting the post-larvae of shrimp with typical syndromes of pale or colorless hepatopancreas and digestive tract, had become an urgent threat to the shrimp farming industry in China (4). In a recent study, a novel *V. parahaemolyticus* ($Vp_{TPD}$) was confirmed as the causative agent of the emerging TPD based on the isolation, identification, and testing of the pathogenic agent, according to the four criteria of Koch's postulates (4). However, the pathogenic mechanism of $Vp_{TPD}$

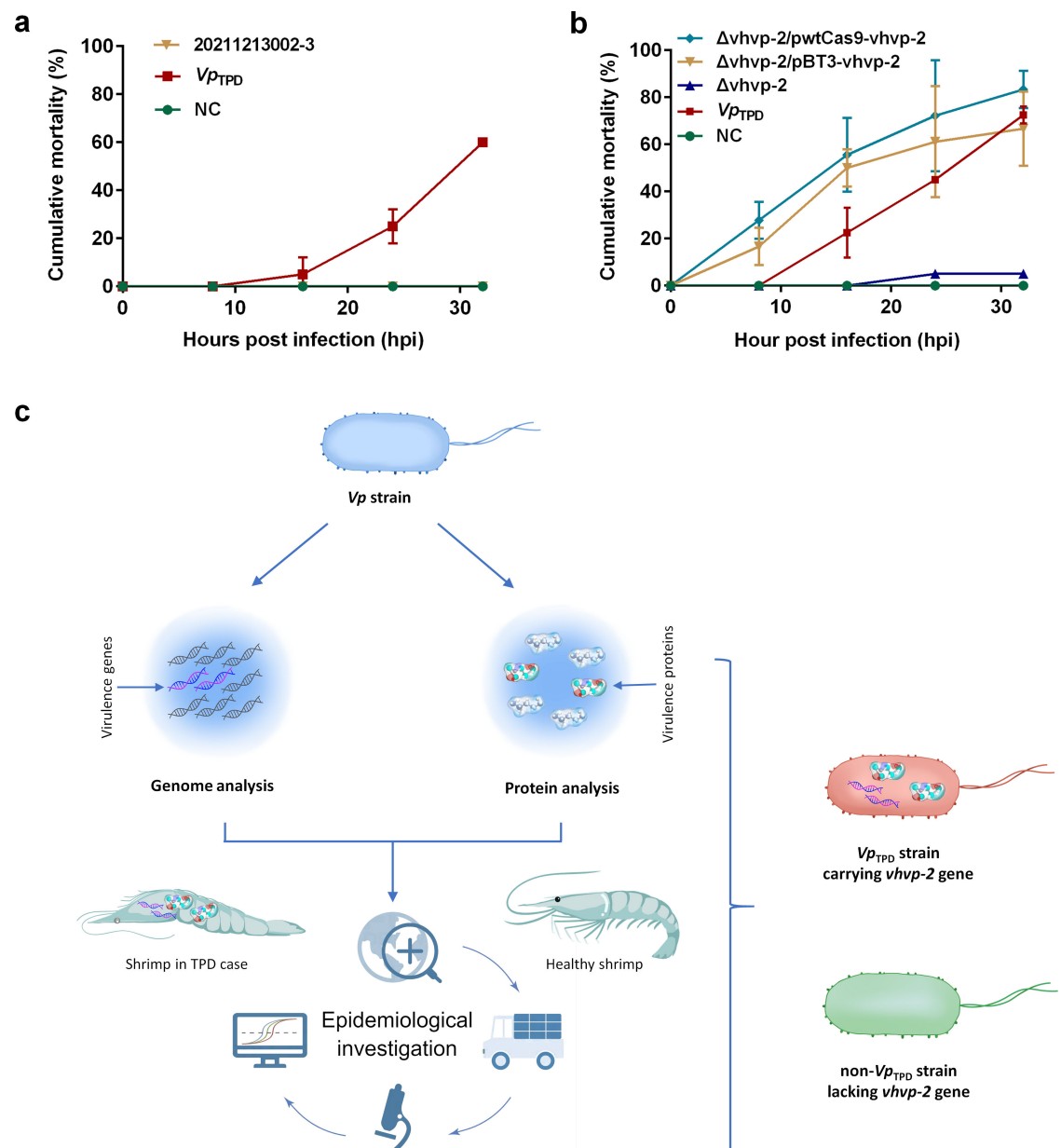

**FIG 5** Identifying the key virulence factor of $Vp_{TPD}$. (a) Cumulative mortality of *Penaeus vannamei* post-larvae immersed in NC, $Vp_{TPD}$, and *Vibrio parahaemolyticus* strain 20211213002-3. *P. vannamei* post-larvae were immersed with the wild-type *Vibrio parahaemolyticus* strain $Vp_{TPD}$ carrying *vhvp-2* gene or the *Vibrio parahaemolyticus* strain 20211213002-3 lacking vhvp-2 gene at the same pathogen dose. Shrimps were monitored daily for mortality. Cumulative shrimp mortality was shown as the average mean and SD of two replicate data for each experimental group. (b) Cumulative mortality of *P. vannamei* post-larvae immersed in NC, $Vp_{TPD}$, Δvhvp-2, Δvhvp-2/pwtCas9-vhvp-2, and Δvhvp-2/pBT3-vhvp-2. The post-larvae of *P. vannamei* were immersed with $Vp_{TPD}$, the *vhvp-2* gene deletion mutant Δvhvp-2, or the *vhvp-2* gene complement strain Δvhvp-2/pwtCas9-vhvp-2 and Δvhvp-2/pBT3-vhvp-2 at the same pathogen dose. Shrimps were monitored daily for mortality. Cumulative mortality of shrimp was shown as the average mean and SD of three replicate data for each experimental group. (c) Schematic of the procedures used to identify the key virulence factor of $Vp_{TPD}$.

was not fully understood yet, which limited the effective prevention and control of $Vp_{TPD}$ in actual shrimp farming practice. In this study, we carried out in-depth investigations, including immersion challenge tests, mass spectrometry analysis, histopathological analysis, and comparative genomic analysis, in order to identify the specific virulence factors of $Vp_{TPD}$ causing translucent post-larvae disease in *P. vannamei*. The results showed that novel toxin protein, designated as VHVP-2 (MW >100 kDa), containing the

conserved domains of *Salmonella* virulence plasmid protein, insecticidal toxin complex protein, and neuraminidase, was the key virulence factor of $Vp_{TPD}$ (Fig. 5c).

The immersion challenge tests in the present study showed that a specific protein fraction, in which MW >100 kDa from the lysate of $Vp_{TPD}$, could cause a similar lethality to the shrimp post-larvae as the live pure culture of $Vp_{TPD}$. This result initially indicated that the virulence factor of $Vp_{TPD}$ should be in the protein fraction with MW >100 kDa. In addition, the supernatant of $Vp_{TPD}$ culture did not show any significant virulent effects to shrimp post-larvae in comparison to that of PBS in our experimental challenge tests, which indicated that the key virulent protein of $Vp_{TPD}$ was likely not secretory (data not shown) under the cultured condition of the present study. Previous studies reported that SDS-PAGE and mass spectrometry analysis have been widely applied for the identification of bacteria virulent proteins with different sizes. For example, PirA- and PirB-like proteins of 13 kDa and 50 kDa were identified as the virulence factor of $Vp_{AHPND}$ by SDS-PAGE and LC-MS/MS in the investigation of the shrimp pathogen of AHPND (5). In addition, proteomic analysis using LC-MS/MS was also applied to elucidate the pathogenesis of *Edwardsiella tarda* (6) as well as to map lysine acetylation sites in revealing their virulent role in *V. alginolyticus* (7). Similarly, SDS-PAGE and mass spectrometry analysis were also successfully applied in the present study to identify virulent proteins of $Vp_{TPD}$. Among three major protein fragments (MW >100 kDa) in $Vp_{TPD}$ based on SDS-PAGE analysis, two of them ($Vp_{TPD}$_4-2-1 and $Vp_{TPD}$_4-2-2) were found to share high sequence similarity with the known virulence factor by mass spectrometry analysis, and they were determined to be the candidate virulence factor of $Vp_{TPD}$. Interestingly, the highly homologous proteins of the candidate virulence protein of $Vp_{TPD}$, including WP_269169668.1 and APX09935.1, were submitted to NCBI GenBank by other researchers in 2017, suggesting that the strains carrying them should have started to spread in some areas of the world before 2017 or earlier. Regarding the bacteria strains carrying the homologous virulent protein, their distribution, transmission mode, and their pathogenic effects to aquatic animals are worthy of further investigation.

It has been well recognized that the methodologies, such as genome sequencing, comparative genomic analysis, and proteomic analysis, play crucial roles in investigating the pathogenesis of AHPND in shrimp (8–11). For example, genome sequence analysis was used to reveal two virulence genes *(pirA-* and *pirB-*like) *of* $Vp_{AHPND}$ in the plasmid pVA1 of $Vp_{AHPND}$ (8, 10, 12, 13), and proteomic analysis confirmed the pir toxin-like proteins encoded by the two genes (5). Moreover, comparative genome analysis further addressed that the virulence genes carrying the transferable plasmids not only exist in $Vp_{AHPND}$ but also in other non-*V. parahaemolyticus* AHPND strains and also contribute to its pathogenesis (13–15). For example, a draft genome sequence showed that a *V. harveyi* isolate could cause AHPND in shrimp in northern Vietnam (16). Studies showed that plasmid-mediated interspecies transfer of the hazard genes could have occurred in different *Vibrio* species, including *V. parahaemolyticus*, *V. campbellii*, and *V. owensii* (17, 18) and that conjugative transfer of the AHPND-causing pVA1-type plasmid carrying the hazard genes is mediated by a novel self-encoded type IV secretion system (19). Such methodologies provide direct guidelines for uncovering the virulence factors and pathogenic mechanism of $Vp_{TPD}$. Our current study based on the abovementioned methods demonstrated that the plasmid containing the virulent *vhvp* gene of $Vp_{TPD}$ also carried *traG, traE, traB, traC,* and other binding transfer-related genes (data not shown), indicating that the virulence gene of *vhvp* in $Vp_{TPD}$ might be able to transfer via conjugation among different *Vibrio* species. Our recent nationwide epidemiological surveys (the data are not shown) also revealed that the *vhvp* gene was identified in a variety of dominant *Vibrio* species, including *V. natriformis, V. Campbellii, and V. alginolyticus*, which were isolated from diseased shrimps with typical TPD symptoms. Our findings suggest that TPD was caused by different pathogens carrying the same transferable *vhvp* genes, and therefore we need to pay more attention on the $Vp_{TPD}$ for its higher risk of horizontal transmission.

The conjoint analysis of mass spectrometry, complete genome sequencing, and comparative genome of $Vp_{TPD}$ indicated the amino acid sequence of the two potential virulence factors of $Vp_{TPD}$ ($Vp_{TPD}$_4-2-1 and $Vp_{TPD}$_4-2-2) annotated by mass spectrometry analysis shared 100% and 99.49% identity with the deduced protein sequence of the two potential candidate virulence genes, GE005140 and GE005139, in the $Vp_{TPD}$ plasmid. Thus, the $Vp_{TPD}$_4-2-1 and $Vp_{TPD}$_4-2-2 proteins were determined to be the putative key toxins of $Vp_{TPD}$, and the genes of GE005140 (*vhvp-1*) and GE005139 (*vhvp-2*) in the 187,791 bp plasmid were identified as the putative virulence genes of $Vp_{TPD}$. $Vp_{TPD}$ *vhvp-1* genes were predicted to encode the four conserved protein domains including *SpvA* (GenBank: CL21676), neuraminidase (GenBank: pfam18413), TcA receptor binding (GenBank: CL139842.1), and *TcA/TcB* super family (GenBank: cl39627), and $Vp_{TPD}$ *vhvp-2* genes were predicted to encode the three conserved domains including *SpvB* (GenBank: pfam03534), *TcdB*_toxin_midC (GenBank: pfam12255), and *TcdB*_toxin_midN superfamily (GenBank: cl13663).

The *Spv* protein has been identified as one of the most important virulence factors of *Salmonella* (20–22), and the *SpvB* protein was reported to act as an intracellular toxin that covalently modified monomeric actin, leading to loss of F-actin filaments and depolymerization of the cytoskeleton in *Salmonella*-infected human macrophages (23–25). The C-terminal domain of *SpvB* was reported to contain ADP-ribosyl transferase activity, which modifies G-actin monomers and prevents their polymerization into F-actin filaments (25, 26), and *SpvB* has been shown to increase cell damage mainly through its F-actin depolymerization-associated function and induction of apoptotic cell death (27-28, 29). The insecticidal toxin complex protein was composed of several subunits including *TcA*, *TcB*, *TcC*, and *TcD*; TcA facilitates receptor-toxin interaction and membrane permeation; TcB and TcC form a toxin-encapsulating cocoon (30–32). It has been reported that the *TcdB* toxin may act synergistically with another glycosylating toxin, TcdA. First, TcdA acts to disrupted epithelial integrity and then allows *TcdB* to enter and mediate toxic effects within the host (31, 33, 34). In addition, *TcdB* has been shown to disrupt epithelial integrity and cause tissue damage in human colon explants (35, 36). During infection, it is likely that *TcdB* first engages NECTIN3 and frizzled proteins to enter and intoxicate the colonic epithelium. Following epithelial damage or loss of tight junctions, the toxin could gain access to CSPG4 in the subepithelial myofibroblasts, causing further mucosal damage (37–39). A previous study on $Vp_{TPD}$ showed that necrosis and sloughing of the epithelial cells occurred in the hepatopancreatic tubules and midgut of naturally infected or immersion-challenged *P. vannamei* post-larval individuals (4). Correspondingly, the same histopathological changes, including necrosis and sloughing of the hepatopancreatic and enteric epithelial cells, also occurred in *P. vannamei* post-larvae from the live $Vp_{TPD}$-challenged group and the >100 kDa proteins of $Vp_{TPD}$-challenged group in the present study. That is, the above pathological changes in the TPD-affected shrimp individuals were consistent with the known pathological characteristics induced by the predicted novel virulence gene *vhvp-2*, encoding the domains of Spv plasmid toxin and Tc toxins. The epidemiological studies indicated that the *vhvp-2* gene was only present in the diseased shrimps with typical TPD syndromes. Moreover, experiments of deletion and complement mutants of the *vhvp-2* gene in $Vp_{TPD}$ further confirmed that the *vhvp-2* gene plays a key role in the realization of $Vp_{TPD}$ virulence. Meanwhile, the results of the epidemiological investigation and challenge test indicated that the *V. parahaemolyticus* isolate carrying only the vhvp-1 gene and lacking vhvp-2 gene could not cause mortality of experimental *P. vannamei* post-larvae. All the abovementioned results indicated that *vhvp-2* was the key virulence gene of $Vp_{TPD}$ in *P. vannamei*. The functional mechanism of the virulence factor VHVP-2 in causing the shedding of intestinal epithelial cells of $Vp_{TPD}$-infected shrimp deserves further investigation.

*Salmonella* infection (salmonellosis) is a common bacterial disease that affects intestinal tract of animals and humans (40), and the most frequent infection route in humans is through consuming contaminated water or foods (41, 42). Recent reports have

shown that farmed shrimps may serve as potential reservoirs and carriers of *Salmonella* bacteria and, therefore, pose a potential risk to public health (43–47). The present study showed that $Vp_{TPD}$ becomes lethally virulent to shrimp post-larvae because it acquired *vhvp-2* gene encoding the domain of *Salmonella* virulence plasmid 28.1 kDa A protein and 65 kDa B protein (*SpvB*). Meanwhile, these results suggested that the shrimp infected by $Vp_{TPD}$ could pose potential risks to public health as well as the other farmed or wild animals via spreading the virulent *vhvp-2* gene in the aquatic environment.

In summary, we preliminarily demonstrated that a novel virulence protein, VHVP-2, was the key toxin of $Vp_{TPD}$, and it was encoded by *vhvp-2* gene located on a 187,892-bp plasmid of the $Vp_{TPD}$ genome. This means that the opportunistic pathogen *V. parahaemolyticus* becomes lethally virulent to shrimp post-larvae by acquiring the virulence factor of VHVP-2. In addition, this study established a PCR detection method of $Vp_{TPD}$ for early warning of TPD. These results proved new insights into the pathogenic mechanism of $Vp_{TPD}$ and provided the first molecular detection method for $Vp_{TPD}$. The present study would be helpful for further investigation of $Vp_{TPD}$ in terms of its diagnostic technique and pathogenic mechanism, as well as for the prevention and control of TPD.

## MATERIALS AND METHODS

### Experiment shrimp

The specific pathogen-free *P. vanmamei* post-larvae (PL$_3$, body length 4–6 mm) were collected from the Haixingnong Shrimp Breeding Northern Base of BLUMP Seed Industry Technology Co., Ltd in Weifang, Shandong Province. *P. vanmamei* post-larvae were acclimated to the laboratory conditions for 2 days in 10 L glass tanks with continuous aeration (at 24°C, salinity 26 ± 3 g/L), fed three times a day with pelleted commercial feed, and then used for the challenge test.

### Bacterial strains and growth conditions

*V. parahaemolyticus* of $Vp_{TPD}$ (*Vp*-JS20200428004-2) was isolated from moribund *P. vannamei* suffering from translucent post-larvae disease and stored in 15% (vol/vol) glycerol tubes at −80°C in the authors' laboratory (4). The strain was inoculated into tryptic soy broth tubes (Land Bridge Technology, Beijing, China), supplemented with 2% NaCl, and incubated for 12 h at 28°C, 200 rpm (shaking). *E. coli* strains were obtained from the American Type Culture Collection (ATCC) and grown in Luria-Bertani broth medium at 37°C. Ampicillin, kanamycin, and chloramphenicol concentrations were supplemented at 100, 50, and 34 µg/mL, respectively. The bacterial strains in this study were listed in Table 1. The two *Escherichia coli* strains DH5α λpir and S17–1 λpir were provided by professor Qiyao Wang from East China University of Science and Technology.

### Inactivation of $Vp_{TPD}$

Following the abovementioned steps, both of the lysate protein extracts by ultrasonic disruption of $Vp_{TPD}$ + U and the upper filtrate of lysate protein extract by ultrasonic

TABLE 1 The bacterial strains used in this study

| Strains | Temperature | Source or reference |
| --- | --- | --- |
| *V. parahaemolyticus* strains | | |
| $Vp_{TPD}$ (*Vp*-JS20200428004-2) | 28°C | Zou et al. (4) |
| Δvhvp-2 | 28°C | This study |
| Δvhvp-2/pwtCas9-vhvp-2 | 28°C | This study |
| Δvhvp-2/pBT3-vhvp-2 | 28°C | This study |
| *Escherichia coli* strains | | |
| DH5α λpir | 37°C | Ma et al. (48) |
| S17–1 λpir | 37°C | Ma et al. (48) |

disruption of $Vp_{TPD}$ + U (MW >100 kDa) were prepared. After being filtered through a 0.22 µm pore size syringe filter, the liquid of lysate protein extract by ultrasonic disruption of $Vp_{TPD}$ + U was treated by heating at 65°C for 45 min and designed as the group of $Vp_{TPD}$ + U and H. The upper filtrate portion of $Vp_{TPD}$ + U (>100 kDa) with the same heat treatment was used as the group of $Vp_{TPD}$ + U and H (>100 kDa). To determine the inactivation effect of different inactivation methods on $Vp_{TPD}$, the viable bacteria in pure cultured $Vp_{TPD}$ ($Vp_{TPD}$), the inactivated $Vp_{TPD}$ treated by ultrasonic disruption treatment ($Vp_{TPD}$ + U), and the inactivated $Vp_{TPD}$ treated by ultrasonic disruption and pasteurization ($Vp_{TPD}$ + U and H) were investigated by using plate-spreading technique.

## Preparation of $Vp_{TPD}$ protein fragments with different molecular weights

The pure culture of $Vp_{TPD}$ was centrifuged at 6,000 rpm for 10 min, the pellet was washed twice with 1× PBS and then resuspended in 1× PBS. The concentration of $Vp_{TPD}$ was adjusted to 1.0 of $OD_{600}$ (approximately equivalent to $10^9$ CFU/mL) using a microplate reader, and the accurate concentration of $Vp_{TPD}$ was then confirmed by plate colony counting method. The $Vp_{TPD}$ preparation was used as live $Vp_{TPD}$ for experimental challenge with immersion method. To obtain different molecular weight proteins of $Vp_{TPD}$, the $Vp_{TPD}$ suspension was disrupted using an ultrasonic homogenizer (Xinzhi, Ningbo, China), and the $Vp_{TPD}$ ultrasonic disruption liquid was then filtered through a 0.22-µm filter to eliminate residuals of $Vp_{TPD}$. The filtrate was then transferred to an ultrafiltration tube with a cut-off molecular weight of 100 kDa and centrifuged at 5,000 × $g$ for 20 min. After centrifugation, the liquid in the upper portion above the filter in the ultrafiltration tube was resuspended and washed, then pooled together as the larger bacterial proteins ($Vp_{TPD}$ + U, MW >100 kDa). The filtrated part of the liquid at the bottom of the ultrafiltration tube was transferred to a new ultrafiltration tube with a cut-off molecular weight of 50 kDa and centrifuged at 7,500 × $g$ for 20 min. Similarly, the top portion of liquid was resuspended and collected as the group of $Vp_{TPD}$ + U (MW: 50–100 kDa). Following the same abovementioned protocols, the groups of $Vp_{TPD}$ + U (MW: 30–50 kDa), $Vp_{TPD}$ + U (MW:10–30 kDa), and $Vp_{TPD}$ + U (MW <10 kDa) were prepared by using proper size ultrafiltration tubes, respectively. The procedure and protocol for preparing the different molecular weight proteins of $Vp_{TPD}$ for experimental challenge are shown in Fig. 1a.

## SDS-PAGE analysis of the lysate protein extract of ultrasonic disrupted $Vp_{TPD}$

The lysate protein extracts $Vp_{TPD}$ were analyzed by sodium dodecyl sulfate-polyacrylamide gel electrophoresis using the SurePAGE precast gel (GenScript, Nanjing, China), according to the manufacturer's instructions. Then the gel was visualized after being stained with Coomassie brilliant blue R-250.

## Identification of virulence factor using mass spectrometer

To identify the suspected virulent factor(s) causing TPD, the target bands with a molecular weight greater than 100 kDa were excised from the gel using a sterile scalpel and then subjected to enzymatic hydrolysis of the protein, according to the previous methods (49–51). The digested samples were analyzed through mass spectrometer by using Easy-nLC 1200 (Thermo Scientific, P/N LC140) and Orbitrap Exploris 480 (Thermo Scientific, P/N BRE725533). Following extraction of the mass spectra data with Proteome Discover software, the database was searched using the Sequest search engine. The search parameters were as follows: the database was the *Vibrio* protein library; trypsin digestion, the maximum missed cut was 2; the mass error of the primary precursor ion was 10 ppm; the mass error of the secondary fragment ion was 0.02 Da; methionine (M) oxidation and asparagine (N) deamination were set as variable modifications as described in the previous research (52).

## Genomic DNA preparation and whole-genome sequencing

To clarify the virulence genes encoding the unique virulence protein of $Vp_{TPD}$, the complete genome sequencing and comparative genome analysis of $Vp_{TPD}$ and a non-virulent *V. parahaemolyticus* isolate (ATCC 33847, designed as $Vp_{1616}$) were carried out. The genomic DNAs of $Vp_{TPD}$ and $Vp_{1616}$ were extracted using TIANamp Bacteria DNA Kit (Tiangen Biotech Co., Ltd, Beijing, China) and sequenced using Biomarker Technologies Corporation (Beijing, China) Nanopore Sequencing Technology Platform. The genome sequences of $Vp_{TPD}$ (GenBank: SRR23329176) and $Vp_{1616}$ (GenBank: CP127846 and CP127847) have been deposited to NCBI.

## Genome composition prediction, comments, and comparative genome analysis

Genome composition prediction was mainly divided into three sections including coding regions, non-coding RNA, and repetitive sequences. Repetitive sequences were predicted based on the principle of *de novo* sequencing using Tandem Repeats Finder (TRF) (53). Coding regions in the genome were identified using Glimmer (54), then related genes were predicted. All predicted genes were used as an input for NR, Swiss-Prot, GO, Cluster of Orthologous Groups (COG), EuKaryotic Orthologous Groups (KOG), and Kyoto Encyclopedia of Genes and Genomes (KEGG) databases (55, 56). The deduced proteins from the $Vp_{TPD}$ strain genome were aligned to that of the $Vp_{1616}$ strain genome using Blastp software (v2.5.0).

## Plasmid construction, gene deletion, and complementation

The plasmid pDM4 was provided by professor Qiyao Wang from East China University of Science and Technology. The plasmids pBT3 and pwtCas9 were provided by professor Li Sun from Institute of Oceanology of the Chinese Academy of Sciences. The plasmids, as well as the primers used in detecing and gene deletion, were listed in Tables 2 and 3, respectively. The primers used for the complementation experiment are 5139-C-F (5'-AGAAAAGAATTCAAAAGATCTAAA-GAGGAGAAAGGATCTATGCAAAATATAAATAATCTG-3') and 5139-C-R (5'-GCCTGGAGATCCT-TACTCGAGTCATGCGGTATCGTTTTCATCTTCATTGA-3'), respectively. The primers used for the gene overexpression experiment are 5139-OE-F (5'-GGAGATATACATATGGATAT-CATGCAAAATATAAATAATCTGAAAC-3') and 5139-OE-R (5'-GTGGTGGTGCTCGAGGATATCT-CATGCGGTATCGTTTTC-3'), respectively.

For deletion of the virulence gene (*vhvp*) of $Vp_{TPD}$, pDM4 was used for in-frame deletion as previously described (59). In brief, fragments upstream and downstream of the CDS of the *vhvp-2* gene were amplified and overlapped by PCR and then inserted into pDM4 at the indicated endonuclease sites (Table 3). The deletion mutant, Δvhvp-2, was generated by two-step homologous recombination and verified by PCR and sequencing.

For the construction of the *vhvp-2* gene complement strain, pBT3 was used as previously reported (60). Briefly, the *vhvp-2* gene was cloned with primers 5139 C-F/ 5139 C-R and then inserted into pBT3 at the *Eco*RV site. The resulting plasmid pBT3-vhvp-2 was then electroporated into Δvhvp-2 to yield the complement strain Δvhvp-2/

**TABLE 2** The plasmids used in this study

| Plasmid | Source or reference | Antibiotic used in this study |
|---|---|---|
| pDM4 | Ma et al. (48) | 34 µg/mL chloramphenicol |
| pBT3 | Zhang et al. (57) | 100 µg/mL ampicillin |
| pwtCas9 | Liu et al. (58) | 100 µg/mL ampicillin |
| pDM4*vhvp-2* | This study | 34 µg/mL chloramphenicol |
| pBT3*vhvp-2* | This study | 100 µg/mL ampicillin |
| pwtCas9*vhvp-2* | This study | 100 µg/mL ampicillin |

**TABLE 3** The PCR primers based on the *vhvp* gene for detecting $Vp_{TPD}$[a]

| Targeted gene | Name of primers | Sequence of primers (5′–3′) | Source |
|---|---|---|---|
| *vhvp-1* | $Vp_{TPD}$-*vhvp-1*-F1 | GAGGAGAGTGTTGACCGAAATC | This study |
| | $Vp_{TPD}$-*vhvp-1*-R1 | CTGCGCCAGTAGTAACGATAAG | |
| *vhvp-2* | $Vp_{TPD}$-*vhvp-2*-F1 | GGAGTATTGGTGGGCTGAAA | This study |
| | $Vp_{TPD}$-*vhvp-2*-R1 | GGTAGGCATGGACCGTAAAG | |
| *vhvp-2* | $Vp_{TPD}$- *vhvp-2*-F2 | CTAAGCCTTGGCTCCTGAAA | This study |
| | $Vp_{TPD}$-*vhvp-2*-R2 | CGGTCAGAATATCGGTATCGTAAA | |
| *vhvp-2* | Δ5139upF | TTA<u>GTCGAC</u>GGAGTATTGGTGGGCTGAAA (*Sal*I) | This study |
| | Δ5139upR | TCCATACTCATGGTAGGCATGGACCGTAAAG | |
| | Δ5139doF | CCATGCCTACCATGAGTATGGACTGCCGTTAAG | |
| | Δ5139doR | GGA<u>AGATCT</u>GTCAGCAAAGTATCTCGGTAAGA (*Bgl*II) | |

[a]Underlined nucleotides are restriction sites of the enzymes indicated in the brackets at the ends.

pBT3-vhvp-2. Positive colonies were selected based on the ampicillin resistance, PCR, and sequencing analyses.

For gene overexpression, the PCR product of *vhvp-2* gene was inserted into pwtCas9 bacterial between the *Bgl*II and *Xho*I sites to allow inducible expression of the gene under a tetracycline promoter. The resulting plasmid was introduced into the indicated Δvhvp-2 strain by electroporation. Where appropriate, the expression was induced by the addition of 2 µL of anhydrotetracycline.

## Experimental challenge by immersion

The healthy post-larvae of *P. vannamei* were randomly divided into seven groups (negative control, PBS), positive control (live $Vp_{TPD}$), $Vp_{TPD}$ + U (>100 kDa), $Vp_{TPD}$ + U (50–100 kDa), $Vp_{TPD}$ + U (30–50 kDa), $Vp_{TPD}$ + U (10–30 kDa), $Vp_{TPD}$ + U (<10 kDa), 15 shrimp individuals per group with three replicates for each group. The average body length of the shrimp was 5.5 mm ± 0.2 mm ($n = 10$). The immersion challenge test was performed as previously described by Tran et al. (61) with minor modifications. To determine the pathogenicity of the virulent protein fractions of $Vp_{TPD}$ with different molecular weights to post-larvae of *P. vannamei*, mortalities were monitored every 8 h for 40 h. The moribund shrimps were also collected for histopathological analysis.

*V. parahemolyticus* Δvhvp-2 was cultured as above described and harvested at $OD_{600}$ 1.0. For Δvhvp-2 motility analysis, *P. vannamei* (P$_5$-P$_7$) were randomly divided into three groups (10 shrimps/group). Shrimps from group 1 (NC) were immersed in seawater. The shrimps in the groups 2 and 3 were similarly immersed with 900 µL of $Vp_{TPD}$ and Δvhvp-2, respectively, in 900 mL seawater, and the mortalities of shrimp were recorded for 32 h.

*V. parahemolyticus* $Vp_{TPD}$, Δvhvp-2/pBT3-vhvp-2, and Δvhvp-2/pwtCas9-vhvp-2 were cultured as above described and collected at $OD_{600}$ 0.5. To examine the mortality-inducing capacity of the *V. parahemolyticus* strains, shrimps from four groups (10 shrimps/group) were immersed with $Vp_{TPD}$, Δvhvp-2/pBT3-vhvp-2, and Δvhvp-2/pwtCas9-vhvp-2, and their mortalities were recorded as described above for 32 h.

## Histopathology

The moribund post-larval shrimps were fixed in 4% paraformaldehyde (PFA)–phosphate-buffered saline (PBS) (PFA-PBS) fixative solution for 24 h and then dehydrated through a gradient of ethanol solutions (4). The treated shrimps were then immediately embedded in paraffin. Paraffin sections (3 µm) of each sample were prepared and stained with hematoxylin-eosin (H&E), according to the routine histological procedures described by Lightner (62). The histopathological changes of each sample were visualized and recorded using the Pannoramic MIDI section scanning system (3DHISTECH Ltd, Budapest, Hungary).

## PCR detection of $Vp_{TPD}$

Based on the sequences of $Vp_{TPD}$, three pairs of PCR primers ($Vp_{TPD}$-*vhvp-1*-F1/R1, $Vp_{TPD}$-*vhvp-2*-F1/R1, and $Vp_{TPD}$-*vhvp-2*-F2/*R2*, Table 3) were designed to detect the virulence gene of *vhvp* in the strain of $Vp_{TPD}$. Total genomic DNA extracted from $Vp_{TPD}$ was used as a template for the PCR assay. The reaction mixture contained 1 µL genomic DNA, 10 mM Tris-HCl (pH 8.3), 50 mM KCl, 4 mM MgCl$_2$, 1.5 mM dNTP, 0.4 µM primers ($Vp_{TPD}$-*vhvp-1*-F1/R1, $Vp_{TPD}$-*vhvp-2*-F1/R1, or $Vp_{TPD}$-*vhvp-2*-F2/*R2*), 2.5 U TaKaRa EX *Taq* DNA polymerase (TaKaRa, Dalian, China). The PCR was performed at 94°C for 4 min, followed by 35 cycles of 94°C for 30 s, 58°C for 30 s, and 72°C for 40 s, ending with 72°C for 7 min. The PCR products were then analyzed in a 1.5% agarose gel containing GeneFinder (Bio-V, Xiamen, China). The expected PCR amplicons of the three sets of primers ($Vp_{TPD}$-*vhvp-1*-F/R, $Vp_{TPD}$-*vhvp-2*-F1/R1, and $Vp_{TPD}$-*vhvp-2*-F2/*R2*) were 362 bp, 351 bp, and 303 bp in length, respectively.

## Epidemiological analysis of $Vp_{TPD}$

Shrimp samples were collected from the shrimp farms in Hebei, Shandong, Jiangsu, Hainan, Xinjiang, Hunan, Hubei, and Guangdong provinces of China from April 2020 to 2021. The samples collected from each pond were divided into three parts: the first part was preserved in 95% ethanol for nucleic acid preparation, the second part was used for histopathological assay, and the third part was used for bacterial isolation. The dominant bacterial strains were isolated from the samples, according to the method described previously (4). Shrimp samples were disinfected with 75% alcohol, washed three times with 1× PBS buffer (pH 7.2; Solarbio, Shanghai, China), and then homogenized in 1× PBS buffer. Bacteria in the homogenized liquid were inoculated onto Marine 2216 agar plates for further growth at 28°C. The dominant bacterial strains were further proliferated in Marine 2216 broth and then used for genomic DNA preparation. The PCR method was applied for diagnosis of the presence of $Vp_{TPD}$ in the DNA samples acquired directly from the shrimp tissues or from the bacterial cultures from the shrimp samples.

## Statistical analysis

All experiments were performed in triplicate. Statistical analyses were performed using GraphPad Prism 6 (GraphPad Software, USA). Data were analyzed using Student's *t*-test or one-way ANOVA. Statistical significance was defined as $P < 0.05$.

## ACKNOWLEDGMENTS

The authors would like to thank Professor Li Sun from Institute of Oceanology of the Chinese Academy of Sciences for her generous help in the experiment of complement mutant of $Vp_{TPD}$, thank Professor Qiyao Wang from East China University of Science and Technology for his generous help in the experiment of isogenic mutant of $Vp_{TPD}$, and thank Dr. Xiao Fan and Dr. Yongwei Yan for their generous help in the genomic sequence search and analysis.

This work was supported by the Central Public-interest Scientific Institution Basal Research Fund, CAFS (no. 2020TD39; 2021XT0602), earmarked fund for CARS-48, Project of Species Conservation from the Ministry of Agriculture and Rural Affairs-Marine fisheries resources collection and preservation, Central Public-interest Scientific Institution Basal Research Fund, YSFRI, CAFS (no. 20603022021022 & 20603022023009), and Qingdao Postdoctoral Researcher Applied Research Project.

Q.Z. designed the study. S.L., W.W., T.-T.X., T.J., W.W., and C.W. executed the experiment. K.L. and J.K. supplied the SPF shrimp post-larvae. S.L. and L.X. screened the location of related genes in the genome. G.X. helps to isolate the original $Vp_{TPD}$ strain. S.L. wrote the manuscript. Q.Z. and J.L. revised the manuscript. All authors interpreted the data, critically revised the manuscript for important intellectual contents, and approved the final version.

## AUTHOR AFFILIATIONS

[1]State Key Laboratory of Mariculture Biobreeding and Sustainable Goods, Yellow Sea Fisheries Research Institute, Chinese Academy of Fishery Sciences, Qingdao, Shandong, China

[2]Laboratory for Marine Fisheries Science and Food Production Processes, Laoshan Laboratory, Qingdao, Shandong, China

[3]Key Laboratory of Marine Aquaculture Disease Control, Key Laboratory of Sustainable Development of Marine Fisheries, Ministry of Agriculture and Rural Affairs, Yellow Sea Fisheries Research Institute, Chinese Academy of Fishery Sciences, Qingdao, Shandong, China

[4]School of Sciences and Medicine, Lake Superior State University, Sault Ste. Marie, Michigan, USA

## AUTHOR ORCIDs

Shuang Liu http://orcid.org/0000-0002-2610-4030
Qingli Zhang http://orcid.org/0000-0002-0406-8825

## FUNDING

| Funder | Grant(s) | Author(s) |
| --- | --- | --- |
| MOA \| CAFS \| Central Public-interest Scientific Institution Basal Research Fund, Chinese Academy of Fishery Sciences (Central Public-interest Scientific Institution Basal Research Fund, CAFS) | 2020TD39 | Qingli Zhang |
| MOA \| CAFS \| Central Public-interest Scientific Institution Basal Research Fund, Chinese Academy of Fishery Sciences (Central Public-interest Scientific Institution Basal Research Fund, CAFS) | 2021XT0602 | Qingli Zhang |
| China Agricultural Research System (CARS) | CARS-48 | Qingli Zhang |
| MOA \| CAFS \| Central Public-interest Scientific Institution Basal Research Fund, Chinese Academy of Fishery Sciences (Central Public-interest Scientific Institution Basal Research Fund, CAFS) | 20603022021022 | Qingli Zhang |

## AUTHOR CONTRIBUTIONS

Shuang Liu, Data curation, Formal analysis, Investigation, Methodology, Validation, Visualization, Writing – original draft | Wei Wang, Formal analysis, Investigation, Methodology, Validation | Tianchang Jia, Investigation, Resources, Validation | Lusheng Xin, Investigation, Methodology | Ting-ting Xu, Data curation, Investigation, Methodology | Chong Wang, Investigation, Methodology, Validation | Guosi Xie, Methodology, Resources | Kun Luo, Methodology, Resources | Jun Li, Methodology, Writing – review and editing | Jie Kong, Resources, Supervision | Qingli Zhang, Conceptualization, Data curation, Formal analysis, Funding acquisition, Investigation, Project administration, Resources, Supervision, Validation, Writing – review and editing

## DATA AVAILABILITY

All the high-throughput sequencing has been deposited in GenBank, and the accession numbers have been listed in the context of the paper. The genome sequences of $Vp_{TPD}$ (GenBank: SRR23329176) and $Vp_{1616}$ (GenBank: CP127846 and CP127847) obtained in the present study have been deposited at NCBI.

## ADDITIONAL FILES

The following material is available online.

Open Peer Review

**PEER REVIEW HISTORY (review-history.pdf).** An accounting of the reviewer comments and feedback.

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
