## [Reviewer comments · Microbiology Spectrum]

Microbiology Spectrum

Vibrio parahaemolyticus becomes lethal to post-larvae shrimp via acquiring novel virulence factors

Shuang Liu, Wei Wang, Tianchang Jia, Lu Xin, Tingting Xu, Chong Wang, Guosi Xie, Kun Luo, Jun Li, Jie Kong, and Qingli Zhang

Corresponding Author(s): Qingli Zhang, Yellow Sea Fisheries Research Institute, Chinese Academy of Fishery Sciences

Review Timeline:

Submission Date:	February 3, 2023
Editorial Decision:	March 17, 2023
Revision Received:	June 14, 2023
Editorial Decision:	June 28, 2023
Revision Received:	August 24, 2023
Editorial Decision:	August 29, 2023
Revision Received:	September 5, 2023
Accepted:	September 5, 2023

Editor: Philip Rather

Reviewer(s): The reviewers have opted to remain anonymous.

Transaction Report:

DOI: <https://doi.org/10.1128/spectrum.00492-23>

March 17, 2023

Dr. Qingli Zhang
Yellow Sea Fisheries Research Institute, Chinese Academy of Fishery Sciences
Qingdao
China

Re: Spectrum00492-23 (Vibrio parahaemolyticus becomes deadly virulent to shrimp post-larvae by acquiring novel virulence factors)

Dear Dr. Qingli Zhang:

First, I would like to apologize for the delay in the review of your manuscript, it was difficult to find reviewers. However, I have decided to move forward with the one review I received, as it was very thorough. In addition, I have read your manuscript myself. My specific comments are below and you will also see the comments of reviewer 1. In addition to these comments, your manuscript is in need of careful proofreading by someone proficient in English. Please see that this is carefully done. I also agree with reviewer 1 that complementation analysis needed to be done. My comments are as follows:

1. Line 130: Why was the 4-2-2 protein already given the name "virulence protein" at this stage.
2. Were the gel purified protein of sufficient quantity that they could be tested for larvae killing?
3. Fig. 3d is confusing and difficult to read. Which is VHVP-1 and which is VHVP-2?
4. Line 172-173: Are these PCR primer sets being used to detect the vhvp-1 and vhvp-2 genes or just one gene. Please clarify in this sentence.
5. Section 194-199: please describe the deletions in terms of the actual genes, and then you can indicate what domains were deleted. It is confusing as it is written.
6. Fig. 4a: indicate which gene is vhvp-1 and vhvp-2

Link Not Available

Sincerely,

Philip Rather

Journals Department
Reviewer comments:

Reviewer #1 (Comments for the Author):

Translucent post-larvae disease (TPD) is an emerging disease of *Penaeus vannamei* that has occurred in China. *Vibrio parahaemolyticus* was identified as the causative agent of TPD by Zou et al. (Pathogens 2020, 9, 741). Here, Liu and colleagues extended their previous work and investigated the potential key virulence factors for causing TPD. They identified two proteins, VHVP-1 and VHVP-2, and showed that the deletion of these genes abolished toxicity to shrimp. They also developed the PCR method to detect vhpv-positive strains. Most of the data presented seem convincing to support the conclusions. However, it's disappointing that the presentation is poor and awkward, which detracts from the quality of the manuscript. The text also needs to be edited in English.

Specific comments

1. It is not appropriate to claim to have identified 'virulence factors' without characterization or complementation experiments. At this stage, it should be changed to 'potential virulence factors' throughout the manuscript.
2. The statistical analysis method they employed should be described in the Materials and Methods.
3. The title needs to be edited.
4. I want the authors to be more eloquent in the Introduction. TPD is still unfamiliar to many readers, and differences with AHPND should be included.
5. Ln 86-94: This section is unnecessary. It is unclear why they describe inactivation treatments here. The fractionation procedure can be incorporated into the next section.
6. Fig. 1b: PBS (negative control), live VpTPD (positive control)
7. Fig. 1c: PBS, instead of negative control; live VpTPD, instead of positive control.
8. Fig 2a is unnecessary.
9. Fig. 2b, it is surprising that only a few major protein bands were detected in the >100 kDa fraction of the Vp lysate. This fraction contains a fourth band of around 100 kDa, which should be mentioned. The whole-cell lysate should be included as a control in this Figure or in Fig S1.
10. Figure 2c, to be shown as VpTPD_4-2-2 instead of Vhvp protein.
11. Regarding protein identification by mass spectrometry. Based on their statement, they claim to have identified proteins by mass spectrometry prior to genomic analysis of the strain causing TPD. In this case, a protein with specific peptides can be determined, but it is not identical to the target protein. Should describe accurately. As they note that WP_269169668 and APX09935.1 are already in the GenBank, these proteins and the strains harboring them should be discussed more.
12. Fig. 3a legend: should describe Vp TPD (left) and Vp1616 (right). The style of genome maps should be unified.
13. Are those genome sequences deposited in the public database?
14. Ln 155: "they depended on the same promoter in the plasmid". How did they claim this without any experimental data? This is also the case in the Abstract.
15. Ln 161-166. Fig 3d, not Fig 3c.
16. Fig. 3d: why do they connect two protein sequences and display them as a single sequence? It is difficult to understand. The image quality is so bad.
17. Fig. 4a, Is this vhpv-2? Do SpvB and TcdB_TN mean domains? Clarify. In addition, if images of amplicons are included in the diagram, at least match primer positions and amplicon widths.
18. Fig. 4b. Lane numbers are necessary in this case.
19. Fig. 5a. Indicate the dose and the number of samples.
20. Complementation should be included in Fig. 5a.
21. Fig. 5b is unnecessary (or should be moved to supplemental figures).
22. Table 1. The targeted genes should be vhpv-2, not SpvB and TcdB?
23. Discussion can be reconsidered to be more informative for readers.
24. Ln 221-228, a conventional method is just described.
25. Ln 234-252. Readers would want to know the discussion of the results rather than this introductory information.
26. Are VHVP-1 and VHVP-2 secreted? Have immersion experiments using purified proteins been examined?

Staff Comments:

Preparing Revision Guidelines

- Point-by-point responses to the issues raised by the reviewers in a file named "Response to Reviewers," NOT IN YOUR

COVER LETTER.

- Upload a compare copy of the manuscript (without figures) as a "Marked-Up Manuscript" file.
- Each figure must be uploaded as a separate file, and any multipanel figures must be assembled into one file.
- Manuscript: A .DOC version of the revised manuscript
- Figures: Editable, high-resolution, individual figure files are required at revision, TIFF or EPS files are preferred

Please return the manuscript within 60 days; if you cannot complete the modification within this time period, please contact me. If you do not wish to modify the manuscript and prefer to submit it to another journal, please notify me of your decision immediately so that the manuscript may be formally withdrawn from consideration by Microbiology Spectrum.

Response to Editor

Re: Spectrum00492-23 (*Vibrio parahaemolyticus* becomes deadly virulent to shrimp post-larvae by acquiring novel virulence factors)

Query 1(Q1): Line 130: Why was the 4-2-2 protein already given the name "virulence protein" at this stage.

Response 1 (R1): Thank you very much for your comments. The virulence experiments of different molecular weight proteins of Vp_{TPD} on *Penaeus vannamei* showed that the protein in the range of > 100 kDa has the closest virulence with Vp_{TPD} , indicating that the key virulence proteins of Vp_{TPD} were in the range of > 100 kDa. Three protein bands in the range of > 100 kDa were identified, of which 4-2-1 and 4-2-2 were annotated as proteins related to bacterial virulence. Therefore, the proteins Vp_{TPD_4-2-1} and Vp_{TPD_4-2-2} were identified to be the candidate virulence factors of Vp_{TPD} , and we have changed the "virulence protein" to "candidate virulence protein" in the revised manuscript.

Q2: Were the gel purified protein of sufficient quantity that they could be tested for larvae killing?

R2: The native tertiary structure of the gel-purified protein was theoretically lost during denaturing electrophoresis. The destruction of the tertiary structure of the protein affected the realization of the protein function, so we did not use the gel-purified protein to test the virulence of the putative virulence proteins.

Q3: Fig. 3d is confusing and difficult to read. Which is VHVP-1 and which is VHVP-2?

R3: In the revised Figure 3d, VHVP-1 and VHVP-2 are shown for clarity, respectively.

Q4: Line 172-173: Are these PCR primer sets being used to detect the *vhvp-1* and *vhvp-2* genes or just one gene. Please clarify in this sentence.

R4: Given that the *vhvp* genes was identified to be the putative key virulence gene based on the analysis of comparative genome, bioassay of > 100 kDa protein and mass spectrometry, so the PCR primers were set for detecting the *vhvp-1* and *vhvp-2* genes. And the context in the manuscript was changed.

Q5: Section 194-199: please describe the deletions in terms of the actual genes, and then you can indicate what domains were deleted. It is confusing as it is written.

R5: To investigate the functional significance of the VHVP-2, an isogenic mutant of Vp_{TPD} , $\Delta vhvp-2$, was constructed, in which almost the entire VHVP-2, from the 136th amino acid residue of SpvB to the 946th residue of TcdB_toxin_midC, was deleted markerlessly. And the detailed information of the deletions has been added in the revised manuscript.

Q6: Fig. 4a: indicate which gene is *vhvp-1* and *vhvp-2*

R6: Thank you very much for your comments. We have redrawn the schematic structure of the Vp_{TPD} *vhvp-1* and *vhvp-2* genes and primers targeting them to make it be more clear in the revised manuscript.

Response to Reviewer 1

Specific comments

Query 1(Q1): It is not appropriate to claim to have identified 'virulence factors' without characterization or complementation experiments. At this stage, it should be changed to 'potential virulence factors' throughout the manuscript.

Response 1 (R1): Thank you very much for your comments. The complementation experiments have been completed and the result of the complementation experiments has been included in the revised manuscript.

Q2: The statistical analysis method they employed should be described in the Materials and Methods.

R2: The description of the statistical analysis method has been added to the section of “Materials and Methods” of the revised manuscript.

Q3: The title needs to be edited.

R3: The title has been edited in the revised manuscript.

Q4: I want the authors to be more eloquent in the Introduction. TPD is still unfamiliar to many readers, and differences with AHPND should be included.

R4: The comparative description of TPD versus AHPND has been added to the revised introduction.

Q5: Ln 86-94: This section is unnecessary. It is unclear why they describe inactivation treatments here. The fractionation procedure can be incorporated into the next section.

R5: Here we tried several different ways of inactivating bacteria to screen for the best way to inactivate $V_{p_{TPD}}$. And then we chose the best one to inactivate $V_{p_{TPD}}$ in subsequent infection experiments. We think it is worth keeping this section here to make it easier for the reader to understand.

Q6: Fi. 1b: PBS (negative control), live $V_{p_{TPD}}$ (positive control)

R6: The text has been amended.

Q7: Fig. 1c: PBS, instead of negative control; live $V_{p_{TPD}}$, instead of positive control.

R7: The text has been amended.

Q8: Fig 2a is unnecessary.

R8: This sketch map could help readers to understand this part of the content quickly and efficiently, and we hope to keep it.

Q9: Fig. 2b, it is surprising that only a few major protein bands were detected in the >100 kDa fraction of the V_p lysate. This fraction contains a fourth band of around 100 kDa, which should be mentioned. The whole-cell lysate should be included as a control in this Figure or in Fig S1.

R9: From the electrophoretogram of denaturing SDS-PAGE of the bacterial lysate, there were only 3 proteins larger than 100 kDa, the fourth band in the electrophoretogram of SDS-PAGE was obviously less than 100 kDa in molecular weight. According to the instruction manual of the ultrafiltration membrane we used, the 100 kDa type ultrafiltration membrane can theoretically only intercept the protein that is obviously larger than 100 kDa in molecular weight, that is to say, the $V_{p_{TPD}}$ protein intercepted by ultrafiltration membrane of 100 kDa type is larger than 100 kDa. Therefore, there seems to be no more reason to analyze the fourth band of in the electrophoretogram of SDS-PAGE which is obviously smaller than 100 kDa. In addition,

according to the reviewer's suggestion, the whole-cell lysate has been included as a control in Fig S1 in the revised manuscript.

Q10: Figure 2c, to be shown as VpTPD_4-2-2 instead of Vhvp protein.

R10: Following the reviewer's suggestion, the Vhvp protein in the Figure 2c has been changed to *Vp_{TPD}_4-2-2* in the revised manuscript.

Q11: Regarding protein identification by mass spectrometry. Based on their statement, they claim to have identified proteins by mass spectrometry prior to genomic analysis of the strain causing TPD. In this case, a protein with specific peptides can be determined, but it is not identical to the target protein. Should describe accurately. As they note that WP_269169668 and APX09935.1 are already in the GenBank, these proteins and the strains harboring them should be discussed more.

R11: According to the process of *Vp_{TPD}* virulence gene identification in this study, we added adjunct word in front of the "virulence gene" to make the result description more accurate in the revised manuscript. We used "candidate virulence gene" in the analysis of SDS-PAGE and mass spectrometry, "putative virulence gene" in the analysis of comparative genome analysis, PCR epidemiological analysis, and "key virulence gene" in the analysis of knockout and compensation experiment. Further discussion of high homologous protein of putative virulence protein of *Vp_{TPD}* has been added in the revised manuscript.

Q12: Fig. 3a legend: should describe Vp TPD (left) and Vp1616 (right). The style of genome maps should be unified.

R12: In order to better differentiate between the two strains, *Vp_{TPD}* (left) and *Vp₁₆₁₆* (right), we have used different colors to express and display their genomic circular map. and we hope to keep the pictures of the two stains of *Vp_{TPD}* (left) and *Vp₁₆₁₆* (right) in different colors in the manuscript for the convenience of readers to distinguish them.

Q13: Are those genome sequences deposited in the public database?

R13: The genome sequences has been deposited to NCBI, and the accession number has been included in the revised manuscript.

Q14: Ln 155: "they depended on the same promoter in the plasmid". How did they claim this without any experimental data? This is also the case in the Abstract.

R14: We have amended the sentence according to your suggestion and added "predict" to describe the actual situation of the possibility of sharing a predicted promoter of the virulence factor *vhvp-1* and *vhvp-2* in the genome in the revised manuscript.

Q15: Ln 161-166. Fig 3d, not Fig 3c.

R15: It's been amended in the revised manuscript.

Q16: Fig. 3d: why do they connect two protein sequences and display them as a single sequence? It is difficult to understand. The image quality is so bad.

R16: The Fig. 3d has been changed in the revised manuscript according to your suggestions for better understanding of the two protein sequences.

Q17: Fig. 4a, Is this *vhvp-2*? Do SpvB and TcdB_TN mean domains? Clarify. In addition, if images of amplicons are included in the diagram, at least match primer positions and amplicon widths.

R17: The Fig. 4a has been amended for better understanding in the revised manuscript according to your suggestions. And the primer position and amplified fragment length has also been shown in details in the revised manuscript.

Q18: Fig. 4b. Lane numbers are necessary in this case.

R18: Lane numbers has been added in the revised manuscript according to your suggestion.

Q19: Fig. 5a. Indicate the dose and the number of samples.

R19: the dose and the number of samples has been added in the section of “Materials and Methods” revised manuscript.

Q20: Complementation should be included in Fig. 5a.

R20: The complementary experiment has been done, and the results have been shown in the new Fig.5b in the revised manuscript according to your suggestion.

Q21: Fig. 5b is unnecessary (or should be moved to supplemental figures).

R21 : This sketch map could help readers to understand the content quickly and efficiently, and we have revised the Fig.5b. and hope to keep it in the manuscript.

Q22: Table 1. The targeted genes should be vhvp-2, not SpvB and TcdB?

R22: The names of targeted genes have been corrected in the revised Table 1.

Q23. Discussion can be reconsidered to be more informative for readers.

R23: At your suggestion, the discussion has been revised, particularly to make it more informative for readers.

Q24: Ln 221-228, a conventional method is just described.

R24: We cited this conventional method to explain that the method used in this study is widely accepted and widely used in identifying the newly virulence proteins.

Q25: Ln 234-252. Readers would want to know the discussion of the results rather than this introductory information.

R25: Considering the very similar of the histopathological characteristics of *Vp*_{TPD} infection in the *P. vannamei* are similar to those of acute hepatopancreaticne-crosis disease (AHPND). As the research progress and results of *Vp*_{AHPND} may have important implications of enlightenment for the research of pathogenicity and diversity of *Vp*_{TPD}, so we summarized the research process and results of the diversity of *Vp*_{AHPND} pathogens here. Following your suggestion, the discussion of the results has also been added to the paragraph.

Q26: Are VHVP-1 and VHVP-2 secreted? Have immersion experiments using purified proteins been examined?

R26: VHVP-1 and VHVP-2 were not shown to be secreted as the supernatant after centrifugation of the bacterial solution did not show any virulent to the shrimp post larvae in our infection experiments.

June 28, 2023

Dr. Qingli Zhang
Yellow Sea Fisheries Research Institute, Chinese Academy of Fishery Sciences
Qingdao
China

Re: Spectrum00492-23R1 (Vibrio parahaemolyticus becomes lethal to shrimp post-larvae for acquiring novel virulence factors)

Dear Dr. Qingli Zhang:

Thank you for submitting your manuscript to Microbiology Spectrum. Your manuscript has been re-reviewed by a previous reviewer and there are still many concerns with the revision, all of which need to be addressed. In addition, the reviewer noted in comment 1 that the complementation experiments need to have wild-type, deletion mutant and complemented strain analyzed together, which was not done in Fig. 5b. Both the reviewer and myself still have concerns about various grammatical errors and I suggest that you have it professionally proofread before resubmission.

Link Not Available

Sincerely,

Philip Rather

Journals Department
Reviewer comments:

Reviewer #1 (Comments for the Author):

1. The main criticism is regarding the infection experiment with the complemented strains, which is presented as Figure 5b in the revision. Figure 5a with WT and the deletion strain and Figure 5b with WT and complemented strains are clearly separated. This should be done including the deletion strain for comparison, otherwise it lacks legitimacy.

2. I would like to recommend authors to check carefully before submitting as a professional. To give some examples, Ln 78,

"hepatopancreaticne-crosis"; Ln 191, "V. Owens" and "V. cambes"; Ln 229, remove the comma; Ln 239, remove "that of"; Ln 350, "famed"; Ln 465, "primes"; italicize gene names and *Vibrio* throughout.

3. Ln 98-106. The purpose of this study is to investigate the virulence factors of Vp causing TPD and its epidemiology as stated by the authors. Nevertheless, this paragraph describing inactivation is confusing and wastes space for the reader. The reasons given in the rebuttal letter also do not seem logical and do not support the need for this section. There needs to be a reconsideration of this paragraph or a logical reason for its inclusion in this paper.

4. Ln 112: "100% mortality was reached" to "mortality reached 100%"

5. Ln 116: "infected with <100 kDa proteins". Infecting proteins?

6. Ln 123. Fig 1c, not 1d. Indicate how long the sample was infected with VpTPD.

7. Ln 142. A brief explanation should be provided here as to why VpTPD_4-2-2 (aconitate hydratase B) was excluded from subsequent analyses.

8. Ln 158. What are "deduced candidate virulence proteins I and II"?

9. Ln 168: "depend on the same promoter" How did they predict that those genes depend on the same promoter? Rather, 'operon' is preferred in this case.

10. Ln 202-204. The strain 20211213002-3 carrying vhvp-2 but lacking vhvp-1, appears without explanation. Given that the infection experiment with this strain shown in Fig. S5 is key to demonstrating the contribution of vhvp-2, but not vhvp-1, to virulence, a detailed explanation of how this strain was found should be provided.

11. Fig S5a: What is the PCR target region for?

12. Table S1. It is unclear which primer set is used to determine the plus/minus of VpTPD. The results of PCR using vhvp-1F/R, vhvp-2F/R and vhvp-2F2/2R2 primers should be accurately described.

13. Ln 220: Δ vhvp-2/pBT3-vhvp-2 and Δ vhvp-2/pwtCas9-vhvp-2, instead of Δ vhvp-2c+(P6-5) and Δ vhvp-2c+(P3-2). Also, explain why the inducible vector was used here.

14. Ln 242-244: The claim that VHVP protein is not secreted must be taken with caution. Do these proteins possess a signal peptide? The culture conditions and data are not provided, so the details are unclear, but if these proteins were not detected in the culture supernatant under a single culture condition, this only indicates that they are not secreted under that specific condition. These should be described appropriately.

15. Ln 258-261: "Interestingly ---, indicating that the strains carrying them should have been prevalent in the world for some time". It is unclear whether this claim is valid or just local, without information on the year and location of isolation of these strains.

16. Information about plasmid 2 (other than the presence of tra genes) is useful in considering the origin and evolution of the plasmid carrying this virulence gene and should be discussed further.

17. Ln 293-306. just repeating the results.

18. Ln 307-329. Not just a list of past observations, I would like to see more discussion on the function of VHVP-2 based on the already-known information.

19. Ln 376. Is the VpTPD strain used in this study identical to Vp-JS20200428004-2? It is not known from previous literature [Ref 3] and should be clearly indicated here and in Table 1.

20. Ln 446. Accession numbers should be provided.

21. Ln 471: "The resulting plasmid --- into the indicated Δ vhvp-2+(3-2) strain by electroporation". This description is not appropriate.

22. Ln 502. Make the primer names consistent with those in Table 3.

23. The legend for Fig 1. The infection dose should be also indicated in figure legends.

24. Ln 800: "Schematic structure of --- gene". This description needs to be changed to properly describe the content of the

figure.

25. Ln 811: "at the same pathogen dose". Indicate dose.

26. Fig 1a, 2a and 5c: It is rather a waste of space for the reader to illustrate such common methods, which may be excluded or moved to supplement.

27. Fig 3a. If the authors used different colors for 'the convenience of readers to distinguish them' as in their response letter, the presented style makes unnecessary confusion for the reader. The label VpTPD or Vp1616 is sufficient to distinguish them with a consistent style. Furthermore, VpTPD should be in panel a, and Vp1616 should be in panel b.

28. Fig 3a and b. It would be better if the displayed size of chromosomes and plasmids reflect their actual size.

29. Fig 3d. Indicate as VHVP-1 and VHVP-2 here, not GE005140 and GE005139 in the legend and RF+1 superfamily in the figure. Also, the quality of the figure is still poor. Should be reconstructed using the protein sequence (size indicates bp) and made clear to the reader.

30. Fig 4c: Since Figure 4c is related to Table S1, I would like to suggest that it would be better to separate it from Figure 4 and combine it with the summary of Table S1 results as an independent figure.

Staff Comments:

Preparing Revision Guidelines

Please return the manuscript within 60 days; if you cannot complete the modification within this time period, please contact me. If you do not wish to modify the manuscript and prefer to submit it to another journal, please notify me of your decision immediately so that the manuscript may be formally withdrawn from consideration by Microbiology Spectrum.

Response to Reviewer 1

Specific comments

Query 1(Q1): The main criticism is regarding the infection experiment with the complemented strains, which is presented as Figure 5b in the revision. Figure 5a with WT and the deletion strain and Figure 5b with WT and complemented strains are clearly separated. This should be done including the deletion strain for comparison, otherwise it lacks legitimacy.

Response 1 (R1): Thank you very much for your comments. The challenge tests of the wild-type, deletion mutant and complemented strains has been analyzed together and shown in the revised manuscript.

Q2: I would like to recommend authors to check carefully before submitting as a professional. To give some examples, Ln 78, "hepatopancreatic ne-crosis"; Ln 191, "*V. Owens*" and "*V. cambes*"; Ln 229, remove the comma; Ln 239, remove "that of"; Ln 350, "famed"; Ln 465, "primes"; italicize gene names and *Vibrio* throughout.

R2: The various grammatical errors have been amended carefully.

Q3: Ln 98-106. The purpose of this study is to investigate the virulence factors of *Vp* causing TPD and its epidemiology as stated by the authors. Nevertheless, this paragraph describing inactivation is confusing and wastes space for the reader. The reasons given in the rebuttal letter also do not seem logical and do not support the need for this section. There needs to be a reconsideration of this paragraph or a logical reason for its inclusion in this paper.

R3: Even if the virulence challenge assay is performed using the same amount of (total protein) of initial concentration of live *Vp*_{TPD}, ultrasonically disrupted *Vp*_{TPD} and different molecular weight potential virulence proteins of *Vp*_{TPD}, live *Vp*_{TPD} will continue to proliferate and increase its toxic effect by continuously producing virulence proteins. To eliminate the risk that ultrasonic disruption might not completely inactivate live *Vp*_{TPD}, leading to the possibility that different molecular weights potential virulence proteins might carry residual active *Vp*_{TPD} in subsequent experiments, we first tested the effect of thermal inactivation and ultrasonic disruption to inactivate *Vp*_{TPD}. We have added the explanation in the revised manuscript.

Q4: Ln 112: "100% mortality was reached" to "mortality reached 100%"

R4: The text has been amended.

Q5: Ln 116: "infected with <100 kDa proteins". Infecting proteins?

R5: The text has been amended.

Q6: Ln 123. Fig 1c, not 1d. Indicate how long the sample was infected with *Vp*_{TPD}.

R6: The error of figure number labeling has been amended and the time has been added in the revised manuscript.

Q7: Ln 142. A brief explanation should be provided here as to why *Vp*_{TPD_4-2-2} (aconitate hydratase B) was excluded from subsequent analyses.

R7: Both *Vp*_{TPD_4-2-1} and *Vp*_{TPD_4-2-2} were selected as the candidate virulence factors I and II of *Vp*_{TPD} for further analysis, and *Vp*_{TPD_4-2-3} were excluded from subsequent analyses, as aconitate hydratase B is not a virulence protein according to previous reports. The brief explanation has been added in the revised manuscript.

Q8: Ln 158. What are "deduced candidate virulence proteins I and II"?

R8: The "deduced candidate virulence proteins I and II" has been revised to "deduced candidate virulence factors I and II", which firstly mentioned in the above paragraph in the

revised manuscript.

Q9: Ln 168: "depend on the same promoter" How did they predict that those genes depend on the same promoter? Rather, 'operon' is preferred in this case.

R9: Here we used BPROM [1], which is a classic bacterial sigma70 promoter recognition program used in many publications, to predict the bacterial promoters.

Reference:

[1] V. Solovyev, A Salamov (2011) Automatic Annotation of Microbial Genomes and Metagenomic Sequences. In Metagenomics and its Applications in Agriculture, Biomedicine and Environmental Studies (Ed. R.W. Li), Nova Science Publishers, p. 61-78

Q10: Ln 202-204. The strain 20211213002-3 carrying vhvp-2 but lacking vhvp-1, appears without explanation. Given that the infection experiment with this strain shown in Fig. S5 is key to demonstrating the contribution of vhvp-2, but not vhvp-1, to virulence, a detailed explanation of how this strain was found should be provided.

R10: The *V. parahaemolyticus* isolate 20211213002-3 was isolated as a common *Vibrio* in the epidemiological investigation as description in the M&M. The detailed information has been included in the context and the Supplemental Table1.

Q11: Fig S5a: What is the PCR target region for?

R11: The PCR primer set of *Vp*_{TPD}-*vhvp-1*-F/R, targeting the conserved domain of TcdA, was designed to test the virulence gene of vhvp-1. And this description has been added in the revised figure caption of S5 Fig.

Q12: Table S1. It is unclear which primer set is used to determine the plus/minus of *Vp*_{TPD}. The results of PCR using vhvp-1F/R, vhvp-2F/R and vhvp-2F2/2R2 primers should be accurately described.

R12: Both of the primer sets of *Vp*_{TPD}-*vhvp-1*-F/R *Vp*_{TPD}-*vhvp-2*-F/R have been used to test the potential virulence genes of the collected bacteria isolates, and the results have been added in the revised Table S1. We amended the Table S1 and added note to make this point clear in the revised manuscript.

Q13: Ln 220: Δ vhvp-2/pBT3-vhvp-2 and Δ vhvp-2/pwtCas9-vhvp-2, instead of Δ vhvp-2c+(P6-5) and Δ vhvp-2c+(P3-2). Also, explain why the inducible vector was used here.

R13: According to the reviewer's suggestion, we changed the names in original lines of Line 39, Line 224, Line 449, Line 470, Line 473, Line 717, Line 807, Line 809 in the revised manuscript. For illustration of that the recovery of bacterial pathogenicity after complement has nothing to do with the complement methods, we conducted the complement experiment by using two different complement methods commonly used at present, including the methods based on both of non-inducible[2] and inducible [3] plasmids according to previous reports.

References:

[2] Zhang WW, Sun K, Cheng S, Sun L. Characterization of DegQVh, a serine protease and a protective immunogen from a pathogenic *Vibrio harveyi* strain. *Appl Environ Microbiol.* 2008. 74, 6254–6262.

[3] Liu X, Wang X, Sun B, Sun L. The Involvement of Thiamine Uptake in the Virulence of *Edwardsiella piscicida*. *Pathogens.* 2022. 11(4):464.

Q14: Ln 242-244: The claim that VHVP protein is not secreted must be taken with caution. Do these proteins possess a signal peptide? The culture conditions and data are not provided, so the details are unclear, but if these proteins were not detected in the culture supernatant under a single

culture condition, this only indicates that they are not secreted under that specific condition. These should be described appropriately.

R14: The culture conditions have been shown in the section "Bacterial strains and growth conditions". Your comment is accurate and exact, and so we added description of that the Vp_{TPD} is not secreted under that specific condition in the revised discussion.

Q15: Ln 258-261: "Interestingly ---, indicating that the strains carrying them should have been prevalent in the world for some time". It is unclear whether this claim is valid or just local, without information on the year and location of isolation of these strains.

R15: The detailed information about the strains carrying the highly homologous protein gene has been added, and the context has been amended in the revised manuscript.

Q16: Information about plasmid 2 (other than the presence of tra genes) is useful in considering the origin and evolution of the plasmid carrying this virulence gene and should be discussed further.

R16: Thank you very much for your comments. The origin and evolution of plasmid 2 information is a very interesting issue. In consideration of that the primary concern in the present study is the key virulence factor, we did not discuss the origin and evolution of plasmid 2 in the discussion. The *vhvp* gene was identified in *V. natriformis*, *V. Campbellii* and *V. alginolyticus*, suggesting the high risk of horizontal transmission of the *vhvp* gene, so the presence of tra genes was also discussed in the manuscript.

Q17: Ln 293-306. just repeating the results.

R17: The paragraph has been compressed and refined in the revised manuscript according to your comment.

Q18: Ln 307-329. Not just a list of past observations, I would like to see more discussion on the function of VHVP-2 based on the already-known information.

R18: The context has been revised according to your comment.

Q19: Ln 376. Is the Vp_{TPD} strain used in this study identical to Vp -JS20200428004-2? It is not known from previous literature [Ref 3] and should be clearly indicated here and in Table 1.

R19: Vp_{TPD} strain used in this study is Vp -JS20200428004-2 and it was indicated here and in Table 1.

Q20: Ln 446. Accession numbers should be provided.

R20: Accession numbers have been provided in the revised manuscript.

Q21: Ln 471: "The resulting plasmid --- into the indicated Δ vhvp-2+(3-2) strain by electroporation". This description is not appropriate.

R21: The description has been changed to "The resulting plasmid pwtCas9-vhvp-2 was introduced into Δ vhvp-2 strain by electroporation."

Q22: Ln 502. Make the primer names consistent with those in Table 3.

R22: The primer names have been revised to be consistent with those in Table 3.

Q23: The legend for Fig 1. The infection dose should be also indicated in figure legends.

R23: The infection dose has been also indicated in figure legends.

Q24: Ln 800: "Schematic structure of --- gene". This description needs to be changed to properly describe the content of the figure.

R24: The description has been changed according to your suggestion.

Q25: Ln 811: "at the same pathogen dose". Indicate dose.

R25: The dose has been indicated in the revised title caption.

Q26: Fig 1a, 2a and 5c: It is rather a waste of space for the reader to illustrate such common methods, which may be excluded or moved to supplement.

R26: For many potential readers, especially the shrimp farmers, the illustration of the methods would be very helpful for easier and better understanding of the present study.

Q27: Fig 3a. If the authors used different colors for 'the convenience of readers to distinguish them' as in their response letter, the presented style makes unnecessary confusion for the reader. The label Vp_{TPD} or Vp_{1616} is sufficient to distinguish them with a consistent style. Furthermore, Vp_{TPD} should be in panel a, and Vp_{1616} should be in panel b.

R27: The genomic chromosome figures of Vp_{TPD} and Vp_{1616} have been re-labelled according to your suggestions.

Q28: Fig 3a and b. It would be better if the displayed size of chromosomes and plasmids reflect their actual size.

R28: The size of chromosomes and plasmids has been clearly labelled in the figures. The authors have attempted to adjust the size of figures of chromosomes and plasmids to reflect their actual size, and the layout of the figures is extremely incongruous.

Q29: Fig 3d. Indicate as VHVP-1 and VHVP-2 here, not GE005140 and GE005139 in the legend and RF+1 superfamily in the figure. Also, the quality of the figure is still poor. Should be reconstructed using the protein sequence (size indicates bp) and made clear to the reader.

R29: The Fig3d has been redrawn. The *vhvp-1* and *vhvp-2* have been highlighted according to your suggestions, and GE005140/GE005139 have been added in parentheses as an explanation to show the corresponding virulence genes in the Vp_{TPD} genome.

Q30: Fig 4c: Since Figure 4c is related to Table S1, I would like to suggest that it would be better to separate it from Figure 4 and combine it with the summary of Table S1 results as an independent figure.

R30: Fig. 4c has been redrawn. And the new Fig. 4c includes the map and the summary of Table S1 results as you suggested.

August 29, 2023

Dr. Qingli Zhang
Yellow Sea Fisheries Research Institute, Chinese Academy of Fishery Sciences
Qingdao
China

Re: Spectrum00492-23R2 (Vibrio parahaemolyticus becomes lethal to post-larvae shrimp via acquiring novel virulence factors)

Dear Dr. Qingli Zhang:

Thank you for submitting your manuscript to Microbiology Spectrum. As you will see your paper is very close to acceptance. I have found some additional minor grammatical changes that I feel should be made. Please modify the manuscript along the lines I have recommended below. As these revisions are quite minor, I expect that you should be able to turn in the revised paper very quickly.

Specific comments

Line 29: Remove the word that. The sentence should read...post larvae shrimp challenged with either...

Line 37: deleting should be changed to deleting

Line 39: complement should be changed to complemented

Line 42: the word shed should be changed to sheds

Line 54: change word "of" to the word "to". The sentence should read.....and would be beneficial to the fisheries department...

Lines 100-106. The section within these lines does not make sense (at least to me). Is it possible to simply delete this section and start with line 106? This could now read as: "We first tested the effects of thermal inactivation and ultrasonic disruption to inactivate..... If this doesn't work for you, please clarify this section to be more clear.

When submitting the revised version of your paper, please provide (1) point-by-point responses to the issues raised by the reviewers as file type "Response to Reviewers," not in your cover letter, and (2) a PDF file that indicates the changes from the original submission (by highlighting or underlining the changes) as file type "Marked Up Manuscript - For Review Only". Please use this link to submit your revised manuscript. Detailed instructions on submitting your revised paper are below.

Link Not Available

Sincerely,

Philip Rather

Reviewer comments:

Preparing Revision Guidelines

To submit your modified manuscript, log onto the eJP submission site at <https://spectrum.msubmit.net/cgi-bin/main.plex>. Go to

Author Tasks and click the appropriate manuscript title to begin the revision process. The information that you entered when you first submitted the paper will be displayed. Please update the information as necessary. Here are a few examples of required updates that authors must address:

Please return the manuscript within 60 days; if you cannot complete the modification within this time period, please contact me. If you do not wish to modify the manuscript and prefer to submit it to another journal, please notify me of your decision immediately so that the manuscript may be formally withdrawn from consideration by Microbiology Spectrum.

Response to Reviewer

Specific comments

Line 29: Remove the word that. The sentence should read...post larvae shrimp challenged with either...

Response: The sentence has been changed as you suggested.

Line 37: deleting should be changed to deleting

Response: The word has been amended.

Line 39: complement should be changed to complemented

Response: The word has been amended.

Line 42: the word shed should be changed to sheds

Response: The word has been amended.

Line 54: change word "of" to the word "to". The sentence should read.....and would be beneficial to the fisheries department...

Response: The word has been amended.

Lines 100-106. The section within these lines does not make sense (at least to me). Is it possible to simply delete this section and start with line 106? This could now read as: "We first tested the effects of thermal inactivation and ultrasonic disruption to inactivate..... If this doesn't work for you, please clarify this section to be more clear.

Response: The sentences your mentioned has been deleted according to your suggestion.

September 5, 2023

Dr. Qingli Zhang
Yellow Sea Fisheries Research Institute, Chinese Academy of Fishery Sciences
Qingdao
China

Re: Spectrum00492-23R3 (Vibrio parahaemolyticus becomes lethal to post-larvae shrimp via acquiring novel virulence factors)

Dear Dr. Qingli Zhang:

Your manuscript has been accepted, and I am forwarding it to the ASM Journals Department for publication. You will be notified when your proofs are ready to be viewed.

Sincerely,

Philip Rather
Editor, Microbiology Spectrum
